# WASSERSTEIN-REGULARIZED CONFORMAL PREDICTION UNDER GENERAL DISTRIBUTION SHIFT

**Rui Xu, Sihong Xie**[*]
The Hong Kong University of Science and Technology (Guangzhou)
rxu233@connect.hkust-gz.edu.cn, sihongxie@hkust-gz.edu.cn

**Chao Chen**
Harbin Institute of Technology
cha01nbox@gmail.com

**Yue Sun, Parvathinathan Venkitasubramaniam**
Lehigh University
yus516@lehigh.edu, pav309@lehigh.edu

## ABSTRACT

Conformal prediction yields a prediction set with guaranteed $1 - \alpha$ coverage of the true target under the i.i.d. assumption, which may not hold and lead to a gap between $1 - \alpha$ and the actual coverage. Prior studies bound the gap using total variation distance, which cannot identify the gap changes under distribution shift at a given $\alpha$. Besides, existing methods are mostly limited to covariate shift, while general joint distribution shifts are more common in practice but less researched. In response, we first propose a Wasserstein distance-based upper bound of the coverage gap and analyze the bound using probability measure pushforwards between the shifted joint data and conformal score distributions, enabling a separation of the effect of covariate and concept shifts over the coverage gap. We exploit the separation to design an algorithm based on importance weighting and regularized representation learning (WR-CP) to reduce the Wasserstein bound with a finite-sample error bound. WR-CP achieves a controllable balance between conformal prediction accuracy and efficiency. Experiments on six datasets prove that WR-CP can reduce coverage gaps to $3.2\%$ across different confidence levels and outputs prediction sets $37\%$ smaller than the worst-case approach on average.

## 1 INTRODUCTION

Because of data noise, unobservable factors, and knowledge gaps, stakeholders must also consider prediction uncertainty in machine learning applications, especially in areas such as fintech (Ryu & Ko, 2020), healthcare (Feng et al., 2021), and autonomous driving (Seoni et al., 2023). Conformal prediction (CP) addresses prediction uncertainty by generating a set of possible targets instead of a single prediction (Vovk et al., 2005; Shafer & Vovk, 2007; Angelopoulos & Bates, 2021). We focus on CP in **regression tasks**. With a trained model $h$, CP calculates the difference (*conformal score*) between the predicted and actual target via a score function $s(x, y) = |h(x) - y|$ over some calibration instances. With the empirical $1 - \alpha$ quantile $\tau$ of the conformal scores, the prediction set $C(x)$ of a test input $x$ contains all targets whose scores are smaller than $\tau$. If calibration and test data are independent and identically distributed (i.i.d.), the probability that the prediction set $C(x)$ contains the true target $y$ of $x$ is close to $1 - \alpha$ (i.e. the *coverage guarantee*).

Denote $P_{XY}$ and $Q_{XY}$ the calibration and test distributions, respectively, in space $\mathcal{X} \times \mathcal{Y}$. We assume $y|x \sim N(f_P(x), \varepsilon_P)$ for $(x, y) \sim P_{XY}$ and $y|x \sim N(f_Q(x), \varepsilon_Q)$ for $(x, y) \sim Q_{XY}$. In practice, the i.i.d. assumption can be violated by a joint distribution shift such that $P_{XY} \neq Q_{XY}$, due to a covariate shift ($P_X \neq Q_X$), a concept shift ($f_P \neq f_Q$), or both (Figure 1(a) left) (Kouw & Loog, 2018). With a distribution shift, the coverage guarantee fails, leading to a gap between the probability that $y \in C(x)$ and $1 - \alpha$. Formally, denoting $P_V$ and $Q_V$ the calibration and test conformal score distributions, respectively, the coverage gap is the difference between the cumulative density functions (CDFs) of $P_V$ and $Q_V$ at quantile $\tau$ (Figure 1(a) left). Prior methods are concerned with

---

[*]Corresponding author: Sihong Xie, sihongxie@hkust-gz.edu.cn

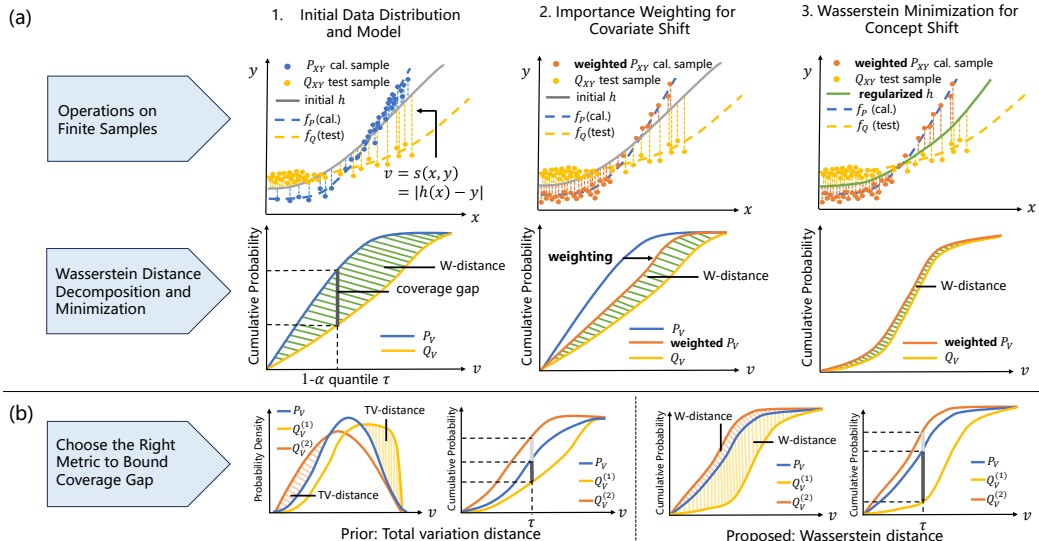

Figure 1: (a) Joint distribution shift can include both covariate shift ($P_X \neq Q_X$) and concept shift ($f_P \neq f_Q$). Coverage gap (Eq. (3)) is the absolute difference in cumulative probabilities of calibration and test conformal scores at the $1 - \alpha$ quantile $\tau$. We address covariate-shift-induced Wasserstein distance by applying importance weighting (Tibshirani et al., 2019) to calibration samples, and further minimize concept-shift-induced Wasserstein distance to obtain accurate and efficient prediction sets; (b) $Q_V^{(1)}$ and $Q_V^{(2)}$ are two distinct test conformal score distributions. Wasserstein distance (Eq. (5)) integrates the vertical gap between two cumulative probability distributions overall *all* quantiles, and is sensitive to coverage gap changes at *any* quantile. Total variation distance fails to indicate coverage gap changes thoroughly as it is agnostic about where two distributions diverge.

the worst-case shifts and passively expand prediction sets as much as possible to meet the coverage guarantee for *any* shifted test distribution, leading to excessively large and inefficient prediction sets (Gendler et al., 2021; Cauchois et al., 2024; Zou & Liu, 2024; Yan et al., 2024). Recent works assume knowledge about the distribution shifts between test and calibration distribution (Barber et al., 2023; Angelopoulos et al., 2022; Colombo, 2024). The knowledge is further embedded as the total variation (TV) distance between conformal score distributions $P_V$ and $Q_V$ to bound and minimize the coverage gap. However, the TV distance ignores where two conformal score distributions differ, while the coverage gap is defined at a specific $\alpha$ and is location-dependent, making TV distance less indicative of coverage gap during model optimization (Figure 1(b) right).

Opposing to TV distance, we adopt Wasserstein distance over the space of probability distributions of conformal score to upper bound the coverage gap under joint distribution shift. Such an upper bound integrates the vertical gap between the CDFs of two conformal score distributions $P_V$ and $Q_V$ and measures the gap at *any* $\alpha$ (Figure 1(b) left), indicating coverage gap at a *given* $\alpha$ for distribution discrepancy minimization and coverage guarantee (Section 3.1, Appendix B). Targeting more effective algorithms specifically for covariate and concept shifts that constitute joint distribution shift, we further penetrate the complex landscape of joint distributional shift. We disentangle the complex dependencies between the Wasserstein upper bound and covariate and concept shifts using a novel pushforwards of probability measure, decomposing the bound into two Wasserstein terms so that the effects of covariate and concept shifts on the coverage gap are independent (Eq. (7)). Theoretical analyses crystalize the link between CP coverage gap, the smoothness of the conformal residue and predictive model, and the amount of covariate and concept shifts (Section 4.1). The decomposition allows representation learning using importance weighting (Tibshirani et al., 2019) that reduces the covariate-shift-induced term, and minimization of the concept-shift-induced term (Figure 1(a)) with finite samples with an empirical error bound (Section 4.2). We proved the effectiveness of the resulting algorithm, Wasserstein-regularized conformal prediction (WR-CP), for multi-source domain generalization where the test distribution is an unknown mixture of training distributions. On six datasets from applications including AI4S (Brooks & Marcolini, 2014), smart transportation (Cui et al., 2019; Guo et al., 2019), and epidemic spread forecasting (Deng et al., 2020), experiments across various $\alpha$ values (0.1 to 0.9) demonstrate that coverage gaps are reduced to 3.2% and the prediction set sizes are 37% smaller than those generated by the worst-case approach on average. Besides, WR-CP allows a smooth balance between prediction coverage and efficiency (Figure 5).

## 2 BACKGROUND AND RELATED WORKS

### 2.1 CONFORMAL PREDICTION

Let $X \in \mathcal{X} \subseteq \mathbb{R}^d$ and $Y \in \mathcal{Y} \subseteq \mathbb{R}$ denote the input and output random variable, respectively. A hypothesis $h : \mathcal{X} \to \mathcal{Y}$ is a model trained to predict target $Y$ from feature $X$. We observe $n$ instances $(X_1, Y_1), ..., (X_n, Y_n)$ from calibration distribution $P_{XY}$. Taking $(x, y)$ as a realization of $(X, Y)$, a score function $s(x, y) : \mathcal{X} \times \mathcal{Y} \to \mathcal{V} \subseteq \mathbb{R}$ quantifies how $(x, y)$ conforms to the model $h$. For regression tasks, typically $s(x, y) = |h(x) - y|$. Split conformal prediction is widely-used, and defines calibration conformal scores $V_i = s(X_i, Y_i)$ for $i = 1, ..., n$ (Papadopoulos et al., 2002). Letting $\tau \in \mathcal{V}$ be the $\lceil (1 - \alpha)(n + 1) \rceil / n$ quantile of $V_1, ..., V_n$, the prediction set of input $X_{n+1}$ is

$$C(X_{n+1}) = \{\hat{y} : s(X_{n+1}, \hat{y}) \leq \tau, \hat{y} \in \mathcal{Y}\}. \tag{1}$$

Consider the instance $(X_{n+1}, Y_{n+1})$ following a test distribution $Q_{XY}$. If the test and calibration instances are i.i.d. (i.e. $P_{XY} = Q_{XY}$), the probability that the true target $Y_{n+1}$ is included in $C(X_{n+1})$ is at least $1 - \alpha$. If calibration conformal scores are almost surely distinct, we can also bound the probability from above by $1 - \alpha + 1/(n + 1)$ (Angelopoulos & Bates, 2021). The bounded probability is called *coverage guanrantee*:

$$\Pr(Y_{n+1} \in C(X_{n+1})) \in [1 - \alpha, 1 - \alpha + 1/(n + 1)). \tag{2}$$

Vovk et al. (2005) proved that the assumption of i.i.d instances can be relaxed to exchangeability of calibration and test instances. With exchangeability, prior CP methods proposed to improve the adaptiveness of prediction set to different test inputs (Romano et al., 2019; 2020; Guan, 2023; Amoukou & Brunel, 2023; Han et al., 2023) and maintain conditional coverage guarantee for sub-populations of the test distribution (Gibbs et al., 2023; Jung et al., 2022; Feldman et al., 2021; Cauchois et al., 2021; Foygel Barber et al., 2021; Stutz et al., 2021; Einbinder et al., 2022b). However, when the assumption is violated so that $P_{XY} \neq Q_{XY}$, coverage guarantee may not hold.

### 2.2 CONFORMAL PREDICTION UNDER DISTRIBUTION SHIFTS

**Covariate shift** $(P_X \neq Q_X)$: Tibshirani et al. (2019) adopted importance weighting by likelihood ratio between $P_X$ and $Q_X$ to satisfy the i.i.d assumption, so coverage is ensured under covariate shift. **Concept shift** $(f_P \neq f_Q)$: Einbinder et al. (2022a); Sesia et al. (2023) addressed CP under concept shift which is represented by label noise.

**Joint distribution shift** $(P_{XY} \neq Q_{XY})$ consists of covariate shift $(P_X \neq Q_X)$ and/or concept shift $(f_P \neq f_Q)$ (Kouw & Loog, 2018). Barber et al. (2023), Angelopoulos et al. (2022), and Angelopoulos & Bates (2021) upper-bound coverage gap via total variation distance, but TV distance cannot identify gap changes at a fixed $\alpha$. To reduce the gap, Gibbs & Candes (2021), Xu & Xie (2021), and Gibbs & Candès (2024) focus on CP under dynamic shift (test distribution changes over time). Meanwhile, some works concentrate on static shift (test distribution unchanged). These works can be categorized into two pipelines. The first pipeline modifies vanilla CP upon a residual-driven model for robust coverage (Gendler et al., 2021; Cauchois et al., 2024; Zou & Liu, 2024). The second pipeline incorporates a conformal-based loss during training to obtain robust and efficient prediction sets (Yan et al., 2024). However, these works treat a joint distribution shift as a whole and adopt a worst-case principle for prediction.

In this work, we explore CP under multi-source domain generalization, which focuses on developing a model that generalizes effectively to **unseen test distributions** by leveraging the data from multiple source distributions (Sagawa et al., 2019; Krueger et al., 2021). A related, yet distinct area is federated CP, which aims to train a model across decentralized data sources to perform well on a known test distribution (typically a uniformly weighted mixture of source distributions) without requiring centralization to ensure privacy. Regarding federated CP, FCP (Lu et al., 2023) and FedCP-QQ (Humbert et al., 2023) aim for a coverage guarantee when the test and calibration samples are exchangeable from the same mixture. When exchangeability does not hold, DP-FedCP (Plassier et al., 2023) addresses scenarios where test samples are drawn from a single source distribution, assuming that only label shifts $(P_Y \neq Q_Y)$ occur among the source distributions. Besides, CP with missing outcomes is studied by Liu et al. (2024) where the samples from the test distribution are accessible. The proposed WR-CP does not consider privacy but works on a more generalized setup: the test samples are drawn from an **unknown random mixture** where both concept and covariate shifts can occur among the source domains.

## 3 METHOD

### 3.1 UPPER-BOUNDING COVERAGE GAP BY WASSERSTEIN DISTANCE

As shown in Figure 1(b), Wasserstein distance can effectively indicate changes in coverage gap across different values of $\alpha$. We formally upper-bound coverage gap via Wasserstein distance. Let $V \in \mathcal{V} \subseteq \mathbb{R}$ be the random variable of conformal score. $P_V$ and $Q_V$ are calibration and test conformal score distributions, respectively. The guarantee in Eq. (2) indicates that $\Pr\left(s(X_{n+1}, Y_{n+1}) \leq \tau\right) \in [1 - \alpha, 1 - \alpha + 1/(n + 1))$. $F_{P_V}$ and $F_{Q_V}$ are CDFs of $P_V$ and $Q_V$, respectively. Under the i.i.d. assumption, $P_V = Q_V$, and thus $F_{Q_V}(\tau) = F_{P_V}(\tau) \in [1 - \alpha, 1 - \alpha + 1/(n + 1))$. However, the assumption can be violated by a joint distribution shift, which may results in $P_V \neq Q_V$. In this case, $F_{P_V}(\tau)$ is still bounded, but $F_{Q_V}(\tau) \neq F_{P_V}(\tau)$. Inadequate coverage renders prediction sets unreliable, while excessive coverage leads to large prediction sets, reducing prediction efficiency, and we define coverage gap as the absolute difference[1]:

$$\text{Coverage gap} = |F_{P_V}(\tau) - F_{Q_V}(\tau)|. \tag{3}$$

**Definition 1** (Kolmogorov Distance). *(Gaunt & Li, 2023) $F_\mu$ and $F_\nu$ are the CDFs of probability measures $\mu$ and $\nu$ on $\mathbb{R}$, respectively. Kolmogorov distance between $\mu$ and $\nu$ is given by $K(\mu, \nu) = \sup_{x \in \mathbb{R}} |F_\mu(x) - F_\nu(x)|$.*

With Definition 1, as $\tau \in \mathcal{V} \subseteq \mathbb{R}$, Eq. (3) is bounded by $K(P_V, Q_V)$:

$$\text{Coverage gap} = |F_{P_V}(\tau) - F_{Q_V}(\tau)| \leq \sup_{v \in \mathcal{V}} |F_{P_V}(v) - F_{Q_V}(v)| = K(P_V, Q_V). \tag{4}$$

**Definition 2** (*p*-Wasserstein Distance). *(Panaretos & Zemel, 2019) Given two probability measures $\mu$ and $\nu$ on a metric space $(\mathcal{X}, c_{\mathcal{X}})$, where $\mathcal{X}$ is a set and $c_{\mathcal{X}}$ is a metric on $\mathcal{X}$, the Wasserstein distance of order $p \geq 1$ between $\mu$ and $\nu$ is*

$$W_p(\mu, \nu) = \inf_{\gamma \in \Gamma(\mu, \nu)} \left( \int_{\mathcal{X} \times \mathcal{X}} c_{\mathcal{X}}(x_1, x_2)^p \, \mathrm{d}\gamma(x_1, x_2) \right)^{1/p}, \tag{5}$$

*where $\Gamma(\mu, \nu)$ is the set of all joint probability measures $\gamma$ on $\mathcal{X} \times \mathcal{X}$ with marginals $\gamma(\mathcal{A} \times \mathcal{X}) = \mu(\mathcal{A})$ and $\gamma(\mathcal{X} \times \mathcal{B}) = \nu(\mathcal{B})$ for all measurable sets $\mathcal{A}, \mathcal{B} \subseteq \mathcal{X}$.*

**Proposition 1.** *(Ross, 2011) If a probability measure $\mu$ in space $\mathbb{R}$ has Lebesgue density bounded by $L$, then for any probability measure $\nu$, $K(\mu, \nu) \leq \sqrt{2LW_1(\mu, \nu)}$.*

In this work, let $W$ denote Wasserstein distance with $p = 1$. Applying Eq. (4) and Proposition 1 with $L$ as the Lebesgue density bound of $P_V$, we can develop an upper bound by

$$\text{Coverage gap} \leq \sqrt{2LW(P_V, Q_V)}. \tag{6}$$

### 3.2 WASSERSTEIN DISTANCE DECOMPOSITION AND MINIMIZATION

In Eq. (6), we show that the Wasserstein distance $W(P_V, Q_V)$ can effectively bound the coverage gap caused by a joint distribution shift. However, it is still not clear how the two components of joint distribution shift, namely, covariate shift in space $\mathcal{X}$ and concept shift in space $\mathcal{Y}$ lead to $W(P_V, Q_V)$ in space $\mathcal{V}$. Besides, we want the quantified contributions amenable to optimization techniques to reduce $W(P_V, Q_V)$. To the best of our knowledge, there is no prior work that suits this need. Therefore, we propose to upper-bound $W(P_V, Q_V)$ with two discrepancy terms due to covariate and concept shifts, and corresponding optimization methods to reduce $W(P_V, Q_V)$ via minimizing the two terms.

**Definition 3** (Pushforward Measure). *If $\mathcal{X}$ and $\mathcal{Y}$ are separate measurable spaces, $\mu$ is a prbability measure on $\mathcal{X}$, and $f : \mathcal{X} \to \mathcal{Y}$ is a measureable function, define the pushforward $f_{\#}\mu$ of $\mu$ through $f$ such that $f_{\#}\mu(\mathcal{A}) = \mu(f^{-1}(\mathcal{A}))$ for all measurable set $\mathcal{A} \subseteq \mathcal{Y}$.*

With Definition 3, we have $P_Y = f_{P\#}P_X$ and $Q_Y = f_{Q\#}Q_X$. Besides, we define $s_P(x) = s(x, f_P(x)) = |h(x) - f_P(x)|$ for $x \sim P_X$, and $s_Q(x) = s(x, f_Q(x)) = |h(x) - f_Q(x)|$ for $x \sim Q_X$, leading to pushforwards of the conformal score $P_V = s_{P\#}P_X$ and $Q_V = s_{Q\#}Q_X$.

---

[1] In this study, we assume that $n$ is sufficiently large for $F_{P_V}(\tau)$ to be approximated as $1 - \alpha$, allowing us to view Eq. (3) as the difference between $F_{Q_V}(\tau)$ and $1 - \alpha$.

To upper-bound $W(P_V, Q_V)$ about conformal scores according to covariate (concept, resp.) shifts in the $\mathcal{X}$ ($\mathcal{Y}$, resp.) space, we introduce a pushforward $Q_{V,s_P} = s_{P\#}Q_X$ on $\mathcal{V}$. Since $P_V$ and $Q_{V,s_P}$ are pushforward measures by the same function $s_P$ from $P_X$ and $Q_X$, respectively, $W(P_V, Q_{V,s_P})$ is a measure of covariate shift ($P_X \neq Q_X$). Also, as $Q_{V,s_P}$ and $Q_V$ are pushforward measures from the same source $Q_X$ by $s_P$ and $s_Q$, respectively, $W(Q_{V,s_P}, Q_V)$ can indicate the extent of concept shift ($f_P \neq f_Q$, and thus $s_P \neq s_Q$). The relationships among the pushforward measures are shown in Figure 2. As Panaretos & Zemel (2019) states, the triangle inequality holds that

$$W(P_V, Q_V) \leq W(P_V, Q_{V,s_P}) + W(Q_{V,s_P}, Q_V). \tag{7}$$

With Eq. (7) bounding $W(P_V, Q_V)$, we design an approach to minimize the upper bound. First, we adopt importance weighting, which weights calibration conformal scores with the likelihood ratio $\mathrm{d}Q_X(x)/\mathrm{d}P_X(x)$. Tibshirani et al. (2019) prove that importance weighting can preserve the coverage guarantee when only a covariate shift occurs. However, existing works do not include the weighting technique when dealing with a joint distribution shift. We prove that importance weighting can minimize covariate-shift-induced Wasserstein distance, $W(P_V, Q_{V,s_P})$, even if a concept shift coincides. Given any measurable set $\mathcal{A} \subseteq \mathcal{X}$, $\mathcal{B} := \{s_P(x) : x \in \mathcal{A}\} \subseteq \mathcal{V}$. With Definition 3,

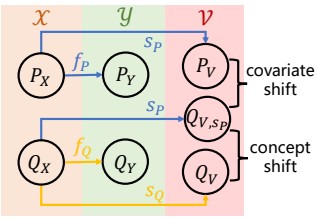

Note: $s_Q(x)=|h(x) - f_Q(x)|$; $s_P(x)=|h(x) - f_P(x)|$.

Figure 2: Pushforward measures.

$$P_V(\mathcal{B}) = \int_{\mathcal{B}} \mathrm{d}P_V(v) = \int_{\mathcal{B}} \mathrm{d}(s_{P\#}P_X)(v) = \int_{\mathcal{A}} \mathrm{d}P_X(x) \quad \overrightarrow{\text{weighting}}$$

$$= \int_{\mathcal{A}} \frac{\mathrm{d}Q_X(x)}{\mathrm{d}P_X(x)} \mathrm{d}P_X(x) = \int_{\mathcal{A}} \mathrm{d}Q_X(x) = \int_{\mathcal{B}} \mathrm{d}(s_{P\#}Q_X)(v) = \int_{\mathcal{B}} \mathrm{d}Q_{V,s_P}(v) = Q_{V,s_P}(\mathcal{B}).$$

Since importance weighting can transform $P_V$ to $Q_{V,s_P}$, $W(P_V, Q_{V,s_P})$ is minimized, and the remaining term in the upper bound in Eq. (7) is the concept-shift-induced component $W(Q_{V,s_P}, Q_V)$. Next, we further minimize it during training, as illustrated in Figure 1. The reasoning behind distinguishing between covariate and concept shifts is elaborated in Appendix C.

## 4 THEORY

### 4.1 UPPER-BOUNDING WASSERSTEIN DISTANCE BY COVARIATE AND CONCEPT SHIFTS

Although Eq. (7) upper bounds for $W(P_V, Q_V)$ by $W(P_V, Q_{V,s_P})$ and $W(Q_{V,s_P}, Q_V)$, it remains unclear how these shifts lead to these terms. Covariate shift can be more accurately quantified by $W(P_X, Q_X)$. Also, with $Q_{Y,f_P} = f_{P\#}Q_X$, $W(Q_{Y,f_P}, Q_Y)$ is a more direct way to measure concept shift by comparing $f_P$ and $f_Q$ based on $Q_X$. Therefore, we further upper-bound the two terms on the right-hand side of Eq. (7) using $W(P_X, Q_X)$ and $W(Q_{Y,f_P}, Q_Y)$. We extend a theorem in Aolaritei et al. (2022), which pushes two probability measures with the same function, while Theorem 1 considers pushing with different functions.

**Theorem 1.** *For probability measures $\mu$ and $\nu$ on metric space $(\mathcal{X}, c_\mathcal{X})$, letting $f, g : \mathcal{X} \to \mathcal{Y}$ be measurable functions, $\mu_f$ and $\nu_g$ on metric space $(\mathcal{Y}, c_\mathcal{Y})$ are pushforwards of $\mu$ and $\nu$ under functions $f$ and $g$, respectively. The Wasserstein distance between $\mu_f$ and $\nu_g$ holds the equivalence:*

$$W(\mu_f, \nu_g) = \inf_{\gamma' \in \Gamma(\mu_f, \nu_g)} \int_{\mathcal{Y} \times \mathcal{Y}} c_\mathcal{Y}(y_1, y_2) \, \mathrm{d}\gamma'(y_1, y_2) = \inf_{\gamma \in \Gamma(\mu, \nu)} \int_{\mathcal{X} \times \mathcal{X}} c_\mathcal{Y}(f(x_1), g(x_2)) \, \mathrm{d}\gamma(x_1, x_2).$$

As $\mathcal{V}, \mathcal{Y} \subseteq \mathbb{R}$, $c_\mathcal{V}(x_1, x_2) = c_\mathcal{Y}(x_1, x_2) = |x_1 - x_2|$, Theorem 1 leads to the following

$$W(Q_{V,s_P}, Q_V) = \inf_{\gamma \in \Gamma(Q_X, Q_X)} \int_{\mathcal{X} \times \mathcal{X}} |s_P(x_1) - s_Q(x_2)| \, \mathrm{d}\gamma(x_1, x_2), \tag{8}$$

$$W(Q_{Y,f_P}, Q_Y) = \inf_{\gamma \in \Gamma(Q_X, Q_X)} \int_{\mathcal{X} \times \mathcal{X}} |f_P(x_1) - f_Q(x_2)| \, \mathrm{d}\gamma(x_1, x_2). \tag{9}$$

Let $\gamma^*$ be the optimal transport plan of $W(Q_{Y,f_P}, Q_Y)$. With $\eta = \max\limits_{x_1, x_2 \in \mathcal{X}} \frac{|s_P(x_1) - s_Q(x_2)|}{|f_P(x_1) - f_Q(x_2)|}$, we have

$$
\begin{aligned}
W(Q_{V,s_P}, Q_V) &\leq \int_{\mathcal{X} \times \mathcal{X}} |s_P(x_1) - s_Q(x_2)| \, \mathrm{d}\gamma^*(x_1, x_2) \\
&\leq \int_{\mathcal{X} \times \mathcal{X}} \eta |f_P(x_1) - f_Q(x_2)| \, \mathrm{d}\gamma^*(x_1, x_2) = \eta W(Q_{Y,f_P}, Q_Y).
\end{aligned}
\tag{10}
$$

In Eq. (10), the first inequality holds as $\gamma^*$ may not be the optimal transport plan of $W(Q_{V,s_P}, Q_V)$, and the second inequality follows the definition of $\eta$. Appendix D shows a geometric intuition of $\eta$.

**Theorem 2.** *For probability measures $\mu$ and $\nu$ on metric space $(\mathcal{X}, c_{\mathcal{X}})$ with a measurable function $f : \mathcal{X} \to \mathcal{Y}$, $\mu_f$ and $\nu_f$ on metric space $(\mathcal{Y}, c_{\mathcal{Y}})$ are the pushforward of $\mu$ and $\nu$ through function $f$, respectively. If $f$ has Lipschitz continuity constant $\kappa$, i.e., $\frac{c_{\mathcal{Y}}(f(x_1), f(x_2))}{c_{\mathcal{X}}(x_1, x_2)} \leq \kappa, \forall x_1, x_2 \in \mathcal{X}$,*

$$
W(\mu_f, \nu_f) \leq \kappa W(\mu, \nu).
\tag{11}
$$

As $\mathcal{X} \subseteq \mathbb{R}^d$, $c_{\mathcal{X}}(x_1, x_2) = \|x_1 - x_2\|_2$. $\kappa$ is the Lipschitz constant of $s_P : \mathcal{X} \to \mathcal{V}$ such that $\frac{|s_P(x_1) - s_P(x_2)|}{\|x_1 - x_2\|_2} \leq \kappa, \forall x_1, x_2 \in \mathcal{X}$. With Theorem 2, as $P_V$ and $Q_{V,s_P}$ are pushforwards of $P_X$ and $Q_X$ through $s_P$, we have

$$
W(P_V, Q_{V,s_P}) \leq \kappa W(P_X, Q_X).
\tag{12}
$$

Plugging Eq. (10) and Eq. (12) into Eq. (7), $W(P_V, Q_V) \leq \kappa W(P_X, Q_X) + \eta W(Q_{Y,f_P}, Q_Y)$. Therefore, by utilizing Eq. (6), we can further bound the coverage gap using the magnitudes of covariate and concept shifts:

$$
\text{Coverage gap} \leq \sqrt{2L \left(\kappa W(P_X, Q_X) + \eta W(Q_{Y,f_P}, Q_Y)\right)}.
\tag{13}
$$

Equation (13) highlights how covariate and concept shifts impact the coverage gap. While the values of $W(P_X, Q_X)$ and $W(Q_{Y,f_P}, Q_Y)$ are inherent properties of given data and cannot be altered, the parameters $\kappa$ and $\eta$ are linked to the model $h$, allowing minimizing $\kappa$ and $\eta$ via optimizing $h$.

## 4.2 Empirical upper bound of coverage gap

In practice, $P_V$ and $Q_V$ are rarely available. Sometimes we may have access to their empirical distributions via the score function $s$, where $\hat{P}_V$ is derived from $n$ calibration samples and $\hat{Q}_V$ is obtained from $m$ test samples. Having the Wasserstein distance between the two empirical distributions $W(\hat{P}_V, \hat{Q}_V)$, we derive the error bound between the empirical form and $W(P_V, Q_V)$ by asymptotic estimation.

**Definition 4** (Upper Wasserstein Dimension). *(Dudley, 1969) Given a set $\mathcal{A} \subseteq \mathcal{X}$, the $\epsilon$-covering number, denoted $\mathcal{N}_\epsilon(\mathcal{A})$, is the minimum $b$ such that $b$ closed balls, $\mathcal{B}_1, ..., \mathcal{B}_b$, of diameter $\epsilon$ achieve $\mathcal{A} \subseteq \cup_{1 \leq i \leq b} \mathcal{B}_i$. For a distribution $\mu$ in $\mathcal{X}$, the $(\epsilon, \zeta)$-dimension is $d_\epsilon(\mu, \zeta) = -\log(\inf\{\mathcal{N}_\epsilon(\mathcal{A}) : \mu(\mathcal{A}) \geq 1 - \zeta\})/\log \epsilon$. The upper Wassersteion dimension with $p = 1$ is*

$$
d_W(\mu) = \inf\{\varphi \in (2, \infty) : \limsup_{\epsilon \to 0} d_\epsilon(\mu, \epsilon^{\frac{\varphi}{\varphi - 2}}) \leq \varphi\}.
\tag{14}
$$

With the definition of upper Wasserstein dimension, Weed & Bach (2019) conducted how an empirical distribution converges to its population by the Wasserstein distance between them.

**Proposition 2.** *(Weed & Bach, 2019) Given a probability measure $\mu$, $\sigma > d_W(\mu)$. If $\hat{\mu}_n$ is an empirical measure corresponding to $n$ i.i.d. samples from $\mu$, $\exists \lambda \in \mathbb{R}$ such that $\mathbb{E}[W(\mu, \hat{\mu}_n)] \leq \lambda n^{-1/\sigma}$. Furthermore, for $t > 0$, $\Pr(W(\mu, \hat{\mu}_n) \geq \mathbb{E}[W(\mu, \hat{\mu}_n)] + t) \leq e^{-2nt^2}$.*

**Theorem 3.** *Given two probability measures $\mu$ and $\nu$, $\sigma_\mu > d_W(\mu)$ and $\sigma_\nu > d_W(\nu)$. $\hat{\mu}_n$ and $\hat{\nu}_m$ are empirical measures corresponding to $n$ i.i.d. samples from $\mu$ and $m$ i.i.d. samples from $\nu$, respectively. For $t_\mu, t_\nu > 0$, $\exists \lambda_\mu, \lambda_\nu \in \mathbb{R}$ with probability at least $(1 - e^{-2nt_\mu^2})(1 - e^{-2mt_\nu^2})$ that*

$$
W(\mu, \nu) \leq W(\hat{\mu}_n, \hat{\nu}_m) + \lambda_\mu n^{-1/\sigma_\mu} + \lambda_\nu m^{-1/\sigma_\nu} + t_\mu + t_\nu.
\tag{15}
$$

Applying Theorem 3 to Eq. (6), we derive an empirical upper bound of coverage gap. Specifically, if $P_V$ has Lebesgue density bounded by $L$, for $t_P, t_Q > 0$, $\sigma_P > d_W(P_V)$, and $\sigma_Q > d_W(Q_V)$, $\exists \lambda_P, \lambda_Q \in \mathbb{R}$ with probability at least $(1 - e^{-2nt_P^2})(1 - e^{-2mt_Q^2})$ that

$$
\text{Coverage gap} \leq \sqrt{2L \left(W(\hat{P}_V, \hat{Q}_V) + \lambda_P n^{-1/\sigma_P} + \lambda_Q m^{-1/\sigma_Q} + t_P + t_Q\right)}.
\tag{16}
$$

## 5  APPLICATION TO MULTI-SOURCE CONFORMAL PREDICTION

In this work, we consider the test distribution to be an unknown random mixture of multiple training distributions, referred to as multi-source domain generalization (Sagawa et al., 2019). As highlighted by Cauchois et al. (2024), achieving 1-$\alpha$ coverage for each of the training distributions ensures that the coverage on test data remains at 1-$\alpha$ if the test distribution is any mixture of the training distributions. We apply the methodology outlined in Section 3 to this scenario, namely multi-source conformal prediction. Given training distributions $D_{XY}^{(i)}$ for $i = 1, .., k$, we require $Q_{XY}$ follows

$$Q_{XY} \in \left\{ \sum_{i=1}^k w_i D_{XY}^{(i)} : w_1, ..., w_k \geq 0, \sum_{i=1}^k w_i = 1 \right\}. \qquad (17)$$

In other words, $Q_{XY}$ is an unknown random mixture of $D_{XY}^{(i)}$ for $i = 1, .., k$. Next, we introduce a surrogate of $W(Q_{V,s_P}, Q_V)$, allowing the minimization of $W(Q_{V,s_P}, Q_V)$ even when the test distribution $Q_{XY}$ is unknown in practice. With the score function $s(x, y)$ and $D_V^{(i)} = s_\# D_{XY}^{(i)}$,

$$Q_V = s_\# Q_{XY} = s_\# \sum_{i=1}^k w_i D_{XY}^{(i)} = \sum_{i=1}^k w_i s_\# D_{XY}^{(i)} = \sum_{i=1}^k w_i D_V^{(i)}. \qquad (18)$$

By marginalizing out $Y$ in Eq. (17), we obtain $Q_X = \sum_{i=1}^k w_i D_X^{(i)}$. Similar to Eq. (18), with score function $s_P(x)$ and $D_{V,s_P}^{(i)} = s_{P\#} D_X^{(i)}$, $Q_{V,s_P} = s_{P\#} Q_X = \sum_{i=1}^k w_i D_{V,s_P}^{(i)}$.

**Theorem 4.** *In space $\mathcal{X} \subseteq \mathbb{R}$, $\nu$ is a mixture distribution of multiple distributions $\nu^{(i)}$, $i = 1, ..., k$, such that $\nu = \sum_{i=1}^k w_i \nu^{(i)}$ with $w_1, ..., w_k \geq 0, \sum_{i=1}^k w_i = 1$. For any distribution $\mu$ on $\mathcal{X}$, Wasserstein distance has the inequality that $W(\mu, \nu) \leq \sum_{i=1}^k w_i W(\mu, \nu^{(i)})$.*

By Theorem 4, $W(Q_{V,s_P}, Q_V) \leq \sum_{i=1}^k w_i W(Q_{V,s_P}, D_V^{(i)}) \leq \sum_{i=1}^k w_i \sum_{i=1}^k w_i W(D_{V,s_P}^{(i)}, D_V^{(i)})$. The inequality offers a surrogate of $W(Q_{V,s_P}, Q_V)$. Even if $Q_{XY}$ is unknown, with uniformly distributed weights, we minimize the expectation of the surrogate with $w_i = 1/k$ for $i = 1, ..., k$: $\min \frac{1}{k} \sum_{i=1}^k W(D_{V,s_P}^{(i)}, D_V^{(i)})$. Besides reducing the coverage gap, we also want smaller prediction errors, so we include empirical risk minimization (ERM) (Vapnik, 1991) during training. Hence, with a loss function $l$ and a parameterized model $h_\theta$, we merge the constant $1/k$ with a hyperparameter $\beta$, and introduce the objective function

$$\min_\theta \sum_{i=1}^k \mathbb{E}_{(x,y) \sim D_{XY}^{(i)}} [l(h_\theta(x), y)] + \beta \sum_{i=1}^k W(D_{V,s_P}^{(i)}, D_V^{(i)}). \qquad (19)$$

We design Wasserstein-regularized Conformal Prediction (WR-CP) to optimize $h_\theta$ by Eq. (19) with finite samples and generate prediction sets with small coverage gaps. $\mathcal{S}_{XY}^{(i)}$ is the sample set drawn from $D_{XY}^{(i)}$ for $i = 1, ..., k$, and $\mathcal{S}_{XY}^P$ is the sample set drawn from $P_{XY}$. $\mathcal{S}_{XY}^Q$ is a test set containing samples from an unknown distribution $Q_{XY}$. Algorithm 1 shows the implementation of WR-CP. Kernel density estimation (KDE) is applied to obtain $\hat{P}_X$, $\hat{D}_X^{(i)}$, and $\hat{Q}_X$ for the calculation of likelihood ratios, whereas $\hat{D}_V^{(i)}$ and $\hat{D}_{V,s_P}^{(i)}$ are estimated as discontinuous, point-wise distributions to ensure differentiability during training. We show the details of distribution estimation in Appendix E. As Algorithm 1 indicates, in the prediction phase, WR-CP follows the inference procedure of importance-weighted conformal prediction (IW-CP) proposed by Tibshirani et al. (2019). When $\beta = 0$, Eq. (19) returns to empirical risk minimization, and thus WR-CP becomes IW-CP.

## 6  EXPERIMENTS

### 6.1  DATASETS AND MODELS

Experiments were conducted on six datasets: (a) the airfoil self-noise dataset (Brooks & Marcolini, 2014); (b) Seattle-loop (Cui et al., 2019), PeMSD4, PeMSD8 (Guo et al., 2019) for traffic speed prediction; (c) Japan-Prefectures, and U.S.-States (Deng et al., 2020) for epidemic spread forecasting. $k = 3$ for the airfoil self-noise dataset, and $k = 10$ for the other five datasets. We conducted 10 sampling trials for each dataset. Within each trails, we sampled $\mathcal{S}_{XY}^{(i)}$ from each subset $i$, for

---

**Algorithm 1** Wasserstein-regularized Conformal Prediction (WR-CP)

---

**Require:** training set $\mathcal{S}_{XY}^{(i)}$ from distribution $D_{XY}^{(i)}$ for $i = 1, ..., k$; calibration set $\mathcal{S}_{XY}^{P}$ from $P_{XY}$; $N$ training epochs; model $h_\theta$; score function $s(x, y) = |h_\theta(x) - y|$; loss function $l$; balancing hyperparameter $\beta$.

---
**Training Phase:**

---

1:  Obtain $\hat{P}_X$ and $\hat{D}_X^{(i)}$ for $i = 1, ..., k$ by kernel density estimation;
2:  **for** $j = 1$ to $N$ **do**
3:      $\mathcal{S}_V^P = \{s(x, y) : (x, y) \in \mathcal{S}_{XY}^P\}$;
4:      **for** $i = 1$ to $k$ **do**
5:          Obtain $\hat{D}_V^{(i)}$ from $\mathcal{S}_V^{(i)} := \{s(x, y) : (x, y) \in \mathcal{S}_{XY}^{(i)}\}$ by point-wise distribution estimation;
6:          Weight all $v \in \mathcal{S}_V^P$ with normalized $\frac{d\hat{D}_X^{(i)}(x)}{d\hat{P}_X(x)}$, where $x$ is the feature that $(x, y) \in \mathcal{S}_{XY}^P, s(x, y) = v$
7:          Obtain $\hat{D}_{V,s_P}^{(i)}$ from the weighted $\mathcal{S}_V^P$ by point-wise distribution estimation;
8:      **end for**
9:      Optimize $h_\theta$ by $\min_\theta \sum_{i=1}^k \mathbb{E}_{(x,y) \in \mathcal{S}_{XY}^{(i)}} [l(h_\theta(x), y)] + \beta \sum_{i=1}^k W(\hat{D}_{V,s_P}^{(i)}, \hat{D}_V^{(i)})$;
10: **end for**

---
**Prediction Phase:**

---

11: Obtain $\hat{Q}_X$ by kernel density estimation;
12: $\mathcal{S}_V^P = \{s(x, y) : (x, y) \in \mathcal{S}_{XY}^P\}$;
13: Weight all $v \in \mathcal{S}_V^P$ with normalized $\frac{d\hat{Q}_X(x)}{d\hat{P}_X(x)}$, where $x$ is the feature that $(x, y) \in \mathcal{S}_{XY}^P, s(x, y) = v$;
14: $\tau = 1 - \alpha$ quantile of the weighted $\mathcal{S}_V^P$;
15: **for** $(x, y) \in \mathcal{S}_{XY}^Q$ **do**
16:     $C(x) = \{\hat{y} : s(x, \hat{y}) \leq \tau, \hat{y} \in \mathcal{Y}\}$;
17: **end for**

---

$i = 1, ..., k$. Given that calibration and training data are commonly assumed to follow the same distribution in CP, we sampled $\mathcal{S}_{XY}^P$ from the union of the $k$ subsets. Additionally, we generated $10k$ test sets for each dataset in every trial. A multi-layer perceptron (MLP) with an architecture of (input dimension, 64, 64, 1) was utilized in all experimental setups to maintain comparison fairness. The detailed information about datasets and sampling procedure is shown in Appendix F.1. The code of our work is released on `https://github.com/rxu0112/WR-CP`.

## 6.2 Correlation between Wasserstein distance and coverage gap

We demonstrated Wasserstein distance can indicate coverage gap changes across $\alpha$ from 0.1 to 0.9 comprehensively, as illustrated in Figure 1(b). Specifically, for each dataset, $h_\theta$ was optimized by empirical risk minimization. Then, we applied vanilla conformal prediction to each test set and calculated the average value of coverage gaps for $\alpha$ values from 0.1 to 0.9. Meanwhile, we also computed the Wasserstein distances between the calibration and each test conformal score distributions. Our findings highlighted a strong positive monotonic relationship between Wasserstein distance and the average value, indicating its sensitivity to coverage gap changes across different $\alpha$.

**Baselines.** Three baseline distance measures were selected. First of all, total variation (TV) distance was chosen as Barber et al. (2023) aimed to use it to bound coverage gap. Besides, Kullback-Leibler (KL)-divergence and expectation differenec ($\Delta\mathbb{E}$) were selected as they are widely applied in domain adaptation researches (Nguyen et al., 2021; Magliacane et al., 2018).

**Metric.** We applied Spearman's coefficient, $-1 \leq r_s \leq 1$ to quantify the monotonic relationship between distance measures and the average coverage gap. The absolute value of the coefficient represents the strength of the correlation. Its sign indicates if a correlation is positive or negative. A **higher** positive $r_s$ means a **stronger** positive monotonic relation. We show the detailed definition of Spearman's coefficient in Appendix F.2.

**Result.** Table 1 presents Spearman's coefficients between distance measures and the average coverage gap across the six datasets, with the standard deviations shown in parentheses. The highest coefficient is bold and the second-highest coefficient is underlined. The result shows that the Wasserstein distance consistently exhibits a high coefficient, suggesting that Wasserstein distance is an effective indicator of the average coverage gap, and establishing it as a suitable optimization metric for maintaining coverage guarantees across various $\alpha$ values.

Table 1: Spearman's coefficients between distance measures and the average coverage gap

| Dataset | Airfoil | PeMSD4 | PeMSD8 | Seattle | U.S. | Japan |
|---------|---------|--------|--------|---------|------|-------|
| $W$ | **0.59** (0.24) | 0.84 (0.03) | **0.90** (0.03) | **0.84** (0.05) | **0.77** (0.06) | **0.57** (0.05) |
| TV | 0.45 (0.16) | **0.88** (0.03) | 0.86 (0.06) | 0.75 (0.09) | 0.67 (0.10) | 0.37 (0.06) |
| KL | 0.40 (0.21) | 0.49 (0.17) | 0.51 (0.09) | 0.45 (0.17) | 0.60 (0.11) | 0.53 (0.05) |
| $\Delta\mathbb{E}$ | 0.55 (0.19) | 0.78 (0.05) | 0.85 (0.04) | 0.71 (0.06) | 0.68 (0.08) | 0.37 (0.09) |

## 6.3 Evaluation of WR-CP in Wasserstein distance minimization

We proved that WR-CP, utilizing importance weighting, can effectively minimize the Wasserstein distances resulting from both concept shift and covariate shift.

**Baselines.** Besides WR-CP, we also conducted vanilla CP and IW-CP on all sampled datasets.

**Metric.** These approaches were compared based on the Wasserstein distance between calibration and test conformal scores. To place greater emphasis on the vertical coverage gap between conformal score CDFs, the distances were normalized to mitigate the impact of varying score scales across datasets, enabling more meaningful comparisons.

**Result.** Figure 3 shows that WR-CP consistently reduces Wasserstein distance. The extent of these reductions is dependent on the value of $\beta$. However, despite the ability to address covariate-shift-induced Wasserstein distance, importance weighting may not always lead to a reduction, as seen in the case of the Seattle-loop dataset. Further explanation of the phenomenon is provided in Appendix C.

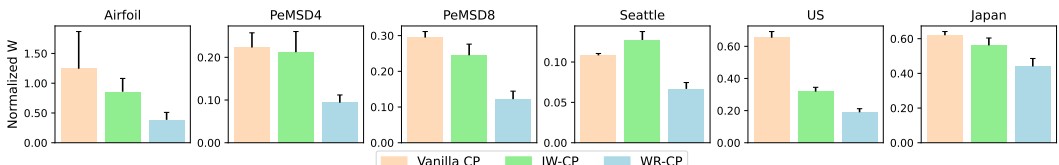

Figure 3: **Comparison of vanilla CP, IW-CP, and WR-CP based on normalized Wasserstein distance between calibration and test conformal scores:** IW-CP can only address the distance caused by covariate shift, while WR-CP reduces the distance from concept shift. The $\beta$ values for the WR-CP method are 9, 11, 9, 10, 13, and 13, respectively.

## 6.4 Robust and efficient prediction sets by WR-CP

We experimentally demonstrated that, compared with prior works, WR-CP is capable of reducing coverage gap without significantly sacrificing prediction efficiency.

**Baselines.** Besides vanilla CP and IW-CP (Tibshirani et al., 2019), conformalized quantile regression (CQR) (Romano et al., 2019) was chosen as a representative method for adaptive CP. We also included the worst-case conformal prediction (WC-CP), which is an implement of the worst-case approach proposed by Gendler et al. (2021); Cauchois et al. (2024); Zou & Liu (2024) in the convex hull setup.

**Metric.** We compared the coverage gaps and sizes of prediction sets generated by WR-CP and baselines as $\alpha$ ranges from 0.1 to 0.9 across all sampled datasets. Prediction sets are **better** when actual coverages are **more concentrated** around $1 - \alpha$ and have **smaller** sizes.

**Result.** With $\alpha = 0.2$, Figure 4 confirms that WR-CP consistently exhibits the most concentrated coverages around $1 - \alpha$ compared to vanilla CP, IW-CP, and CQR across datasets. While WC-CP maintains coverage guarantees under joint distribution shift, it leads to inefficient predictions. In contrast, WR-CP mitigates this inefficiency through smaller set sizes. We show the results with other $\alpha$ values in Appendix F.3. It is important to observe that vanilla CP and and IW-CP always have smaller prediction sets than WR-CP. Since WR-CP is trained with the additional Wasserstein regularization term in Eq. (19), the trade-off inevitably causes an increase in prediction errors, which are proportional to conformal scores. Consequently, methods based on empirical risk minimization,

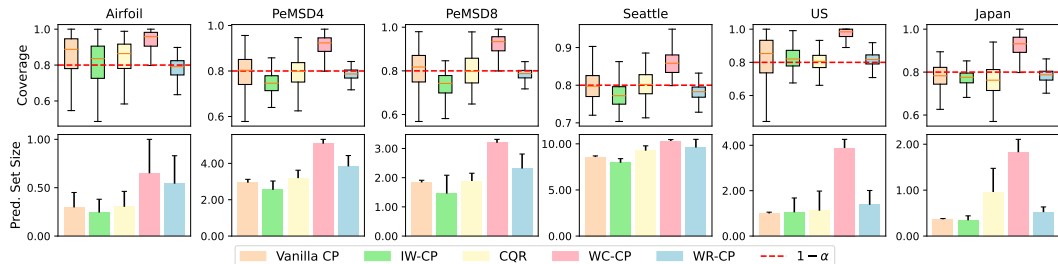

Figure 4: **Coverages and prediction set sizes of WR-CP and baselines with** $\alpha = 0.2$**:** WR-CP makes coverages on test data more concentrated around the $1 - \alpha$ level compared to vanilla CP, IW-CP, and CQR. While WC-CP ensures coverage guarantees, it leads to inefficient predictions due to large set sizes, whereas WR-CP mitigates this inefficiency. The $\beta$ values for the WR-CP method are 4.5, 9, 9, 6, 8, and 20, respectively.

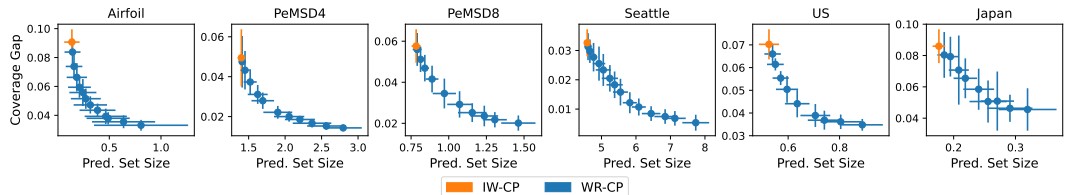

Figure 5: **Pareto fronts of coverage gap and prediction set size obtained from WR-CP with varying** $\beta$**:** WR-CP effectively balances conformal prediction accuracy and efficiency, providing a flexible and customizable solution. When $\beta = 0$, WR-CP returns to IW-CP.

like vanilla CP and IW-CP, tend to yield smaller prediction sets compared to WR-CP due to their lower conformal scores. We further discuss the trade-off in WR-CP in Subsection 6.5. Lastly, we can see IW-CP have worse coverages than vanilla CP on Seattle-loop dataset, reflecting the fact that importance weighting enlarges Wasserstein distance on that dataset in Figure 3.

## 6.5 ABLATION STUDY

As outlined in Eq. (19), WR-CP is regulated by a hyperparameter $\beta$, which governs the trade-off between coverage gap and prediction set size. It is essential to investigate the performance of WR-CP under different $\beta$ values, which are listed in Appendix F.4. To achieve this, we conducted WR-CP on all sampled datasets with varying $\beta$ values. At each $\beta$ value, we calculated the average coverage gap and set size over $\alpha$ from 0.1 to 0.9. Finally, we obtained a Pareto front for each dataset in Figure 5. In particular, when $\beta = 0$, WR-CP reverts back to IW-CP, so we emphasize the outcomes in this scenario as boundary solutions derived from IW-CP. The results indicate that WR-CP allows users to customize the approach based on their preferences for conformal prediction accuracy and efficiency. We further explore whether WR-CP can achieve efficient prediction with a coverage guarantee in Appendix G. The limitations of our study are presented in Appendix H.

## 7 CONCLUSION

In this work, we point out that the coverage gap of conformal prediction under joint distribution shift relies on the distance between the CDFs of calibration and test conformal score distributions. Based on this observation, we propose an upper bound of coverage gap utilizing Wasserstein distance, offering better identifiability of gap changes at different $\alpha$. We conduct a detailed analysis of the bound by utilizing probability measure pushforwards from the shifted joint data distribution to conformal score distributions. This approach allows us to explore the separation of the impact of covariate and concept shifts on the coverage gap. Based on the separation, we design Wasserstein-regularized conformal prediction (WR-CP) via importance weighting and regularized representation learning, which can obtain accurate and efficient prediction sets with controllable balance. The performance of WR-CP is experimentally analyzed with diverse baselines and datasets.

ACKNOWLEDGMENT

Sihong Xie was supported by the Department of Science and Technology of Guangdong Province (Grant No. 2023CX10X079), the National Key R&D Program of China (Grant No. 2023YFF0725001), the Guangzhou-HKUST(GZ) Joint Funding Program (Grant No. 2023A03J0008), and Education Bureau Guangzhou Municipality.

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

# A PROOFS OF THEOREMS

## A.1 PROOF OF THEOREM 1

*Proof.* We define $f \times g$ by $f \times g(x_1, x_2) = (f(x_1), g(x_2)) = (y_1, y_2)$. Let $\mathrm{Id}_{\mathcal{X}}$ be the identity mapping function on $\mathcal{X}$, and let $\pi_i$ be the mapping function to the $i$-th marginal. The proof follows Proposition 3 in the work by Aolaritei et al. (2022).

First, we prove the inclusion that $(f \times g)\#\Gamma(\mu, \nu) \subset \Gamma(f_{\#}\mu, g_{\#}\nu)$. Consider $\gamma \in \Gamma(\mu, \nu)$, so it is equivalent to prove that $(f \times g)_{\#}\gamma \in \Gamma(f_{\#}\mu, g_{\#}\nu)$, which means the marginals of $(f \times g)_{\#}\gamma$ are $f_{\#}\mu$ and $g_{\#}\nu$. For any continuous and bounded function $\phi : \mathcal{Y} \to \mathbb{R}$, we have

$$
\int_{\mathcal{Y} \times \mathcal{Y}} \phi(y_1) \, \mathrm{d}((f \times g)_{\#}\gamma)(y_1, y_2) = \int_{\mathcal{X} \times \mathcal{X}} \phi(f(x_1)) \, \mathrm{d}\gamma(x_1, x_2)
$$
$$
= \int_{\mathcal{X}} \phi(f(x_1)) \, \mathrm{d}\mu(x_1) = \int_{\mathcal{Y}} \phi(y_1) \, \mathrm{d}(f_{\#}\mu)(y_1), \tag{20}
$$

so we obtain $\pi_{1\#}((f \times g)_{\#}\gamma) = f_{\#}\mu$ and similarly derive $\pi_{2\#}((f \times g)_{\#}\gamma) = g_{\#}\nu$.

Secondly, we need to prove $\Gamma(f_{\#}\mu, g_{\#}\nu) \subset (f \times g)\#\Gamma(\mu, \nu)$. With $\gamma' \in \Gamma(f_{\#}\mu, g_{\#}\nu)$, we seek $\gamma \in \Gamma(\mu.\nu)$ such that $(f \times g)_{\#}\gamma = \gamma'$. To do so, let $\gamma_{12} := (\mathrm{Id}_{\mathcal{X}} \times f)_{\#}\mu \in \Gamma(\mu, f_{\#}\mu)$, $\gamma_{23} := \gamma' \in \Gamma(f_{\#}\mu, g_{\#}\nu)$, and $\gamma_{34} := (g \times \mathrm{Id}_{\mathcal{X}})_{\#}\nu \in \Gamma(g_{\#}\nu, \nu)$. As $\pi_{2\#}\gamma_{12} = \pi_{1\#}\gamma_{23} = f_{\#}\mu$, and $\pi_{1\#}\gamma_{34} = \pi_{2\#}\gamma_{23} = g_{\#}\nu$, Santambrogio (2015) ensures a joint probability measure $\bar{\gamma}$ on $\mathcal{X} \times \mathcal{Y} \times \mathcal{Y} \times \mathcal{X}$ satisfying $(\pi_1 \times \pi_2)_{\#}\bar{\gamma} = \gamma_{12}$, $(\pi_2 \times \pi_3)_{\#}\bar{\gamma} = \gamma_{23}$, and $(\pi_3 \times \pi_4)_{\#}\bar{\gamma} = \gamma_{34}$. We demonstrate that $\gamma := (\pi_1 \times \pi_4)_{\#}\bar{\gamma}$ is the probability measure we are seeking. For this, we prove $\gamma \in \Gamma(\mu, \nu)$ with any continuous and bounded function $\phi : \mathcal{X} \to \mathbb{R}$ by

$$
\int_{\mathcal{X} \times \mathcal{X}} \phi(x_i) \, \mathrm{d}\gamma(x_1, x_2) = \int_{\mathcal{X} \times \mathcal{Y} \times \mathcal{Y} \times \mathcal{X}} \phi(x_1) \, \mathrm{d}\bar{\gamma}(x_1, y_1, y_2, x_2)
$$
$$
= \int_{\mathcal{X} \times \mathcal{Y}} \phi(x_1) \, \mathrm{d}\gamma_{12}(x_1, y_1) = \int_{\mathcal{X}} \phi(x_1) \, \mathrm{d}\mu(x_1). \tag{21}
$$

Eq. (21) indicates $\pi_{1\#}\gamma = \mu$. Similarly, we can derive $\pi_{2\#}\gamma = \nu$. As a result, we can prove $(f \times g)_{\#}\gamma = \gamma'$ with any continuous and bounded function $\phi : \mathcal{Y} \times \mathcal{Y} \to \mathbb{R}$ by

$$
\int_{\mathcal{Y} \times \mathcal{Y}} \phi(y_1, y_2) \, \mathrm{d}((f \times g)_{\#}\gamma)(x_1, x_2)
$$
$$
= \int_{\mathcal{X} \times \mathcal{X}} \phi(f(x_1), g(x_2)) \, \mathrm{d}\gamma(x_1, x_2)
$$
$$
= \int_{\mathcal{X} \times \mathcal{Y} \times \mathcal{Y} \times \mathcal{X}} \phi(f(x_1), g(x_2)) \, \mathrm{d}\bar{\gamma}(x_1, y_1, y_2, x_2) \tag{22}
$$
$$
= \int_{\mathcal{X} \times \mathcal{Y} \times \mathcal{Y} \times \mathcal{X}} \phi(y_1, y_2) \, \mathrm{d}\bar{\gamma}(x_1, y_1, y_2, x_2)
$$
$$
= \int_{\mathcal{Y} \times \mathcal{Y}} \phi(y_1, y_2) \, \mathrm{d}\gamma_{23}(y_1, y_2) = \int_{\mathcal{Y} \times \mathcal{Y}} \phi(y_1, y_2) \, \mathrm{d}\gamma'(y_1, y_2).
$$

As $(f \times g)\#\Gamma(\mu, \nu) \subset \Gamma(f_{\#}\mu, g_{\#}\nu)$ and $\Gamma(f_{\#}\mu, g_{\#}\nu) \subset (f \times g)\#\Gamma(\mu, \nu)$, we obtain $(f \times g)\#\Gamma(\mu, \nu) = \Gamma(f_{\#}\mu, g_{\#}\nu)$. Finally, we prove Theorem 1 by

$$
W(\mu_f, \nu_g) = W(f_{\#}\mu, g_{\#}\nu)
$$
$$
= \inf_{\gamma' \in \Gamma(f_{\#}\mu, g_{\#}\nu)} c_{\mathcal{Y}}(y_1, y_2) \, \mathrm{d}\gamma'(y_1, y_2)
$$
$$
= \inf_{\gamma' \in (f \times g)_{\#}\Gamma(\mu, \nu)} \int_{\mathcal{Y} \times \mathcal{Y}} c_{\mathcal{Y}}(y_1, y_2) \, \mathrm{d}\gamma'(y_1, y_2) \tag{23}
$$
$$
= \inf_{\gamma \in \Gamma(\mu, \nu)} \int_{\mathcal{Y} \times \mathcal{Y}} c_{\mathcal{Y}}(y_1, y_2) \, \mathrm{d}((f \times g)_{\#}\gamma)(y_1, y_2)
$$
$$
= \inf_{\gamma \in \Gamma(\mu, \nu)} \int_{\mathcal{Y} \times \mathcal{Y}} c_{\mathcal{Y}}(f(x_1), g(x_2)) \, \mathrm{d}\gamma(y_1, y_2)
$$

$\square$

## A.2 PROOF OF THEOREM 2

*Proof.* Let $\gamma' \in \Gamma(\mu_f, \nu_f)$ be the pushforward of $\gamma \in \Gamma(\mu, \nu)$ via function $f \times f$. We can apply Theorem 1 to $W(\mu_f, \nu_f)$ and obtain

$$W(\mu_f, \nu_f) = \inf_{\gamma \in \Gamma(\mu, \nu)} \int_{\mathcal{X} \times \mathcal{X}} c_{\mathcal{Y}}(f(x_1), f(x_2)) \, d\gamma(x_1, x_2) \,. \tag{24}$$

If the optimal transport plan for $W(\mu, \nu)$ is $\gamma^*$, and $\kappa$ bounds the Lipschitz continuity of $f$, we have

$$
\begin{aligned}
W(\mu_f, \nu_f) &\leq \int_{\mathcal{X} \times \mathcal{X}} c_{\mathcal{Y}}(f(x_1), f(x_2)) \, d\gamma^*(x_1, x_2) \\
&\leq \int_{\mathcal{X} \times \mathcal{X}} \kappa c_{\mathcal{X}}(x_1, x_2) \, d\gamma^*(x_1, x_2) = \kappa W(\mu, \nu).
\end{aligned}
\tag{25}
$$

In Eq. (25), the first inequality holds because $\gamma^*$ may not be the optimal transport plan for $W(\mu_f, \nu_f)$, and the second inequality holds due to the definition of $\kappa$. $\qquad\square$

## A.3 PROOF OF THEOREM 3

*Proof.* As Wasserstein distance satisfies triangle inequality, $W(\mu, \nu)$ and $W(\hat{\mu}_n, \hat{\nu}_m)$ follow

$$W(\mu, \nu) \leq W(\hat{\mu}_n, \mu) + W(\hat{\mu}_n, \nu) \leq W(\hat{\mu}_n, \mu) + W(\hat{\mu}_n, \hat{\nu}_m) + W(\hat{\nu}_m, \nu). \tag{26}$$

Given $\mathbb{E}[W(\mu, \hat{\mu}_n)] \leq \lambda_\mu n^{-1/\sigma_\mu}$ and $\mathbb{E}[W(\nu, \hat{\nu}_m)] \leq \lambda_\nu m^{-1/\sigma_\nu}$ from Proposition 2, with probabilities at least $1 - e^{-2nt_\mu^2}$ and $1 - e^{-2mt_\nu^2}$, respectively, we have

$$W(\mu, \hat{\mu}_n) \leq \lambda_\mu n^{-1/\sigma_\mu} + t_\mu, \; W(\nu, \hat{\nu}_m) \leq \lambda_\nu m^{-1/\sigma_\nu} + t_\nu. \tag{27}$$

It is reasonable to assume the two events in Eq. (27) are independent, so we can apply them to Eq. (26), and thus obtain Eq. (15) with probability at least $(1 - e^{-2nt_\mu^2})(1 - e^{-2mt_\nu^2})$. $\qquad\square$

## A.4 PROOF OF THEOREM 4

*Proof.* We denote $F_\mu$, $F_\nu$, and $F_{\nu^{(i)}}$ the corresponding CDFs of $\mu$, $\nu$, and $\nu^{(i)}$ for $i = 1, ..., k$.

When two distributions are on the real number set $\mathbb{R}$ with Euclidean distance, $W$ of the two distributions equals the area between their CDFs. Therefore, the 1-Wasserstein distance between $\mu$ and $\nu$ is given by

$$W(\mu, \nu) = \int_{\mathcal{X}} |F_\mu(x) - F_\nu(x)| \, dx \,. \tag{28}$$

Since $\nu = \sum_{i=1}^k w_i \nu^{(i)}$, we have $F_\nu(x) = \sum_{i=1}^k w_i F_{\nu^{(i)}}(x)$. As $\nu$, $\nu^{(i)}$, and $\mu$ are defnded on $\mathcal{X} \subseteq \mathbb{R}$, we can derive

$$
\begin{aligned}
W(\mu, \nu) &= \int_{\mathcal{X}} |F_\mu(x) - F_\nu(x)| \, dx = \int_{\mathcal{X}} \left| F_\mu(x) - \sum_{i=1}^k w_i F_{\nu^{(i)}}(x) \right| dx \\
&= \int_{\mathcal{X}} \left| \sum_{i=1}^k w_i F_\mu(x) - \sum_{i=1}^k w_i F_{\nu^{(i)}}(x) \right| dx = \int_{\mathcal{X}} \left| \sum_{i=1}^k w_i \left( F_\mu(x) - F_{\nu^{(i)}}(x) \right) \right| dx \\
&\leq \int_{\mathcal{X}} \sum_{i=1}^k w_i |F_\mu(x) - F_{\nu^{(i)}}(x)| \, dx = \sum_{i=1}^k w_i \int_{\mathcal{X}} |F_\mu(x) - F_{\nu^{(i)}}(x)| \, dx \\
&= \sum_{i=1}^k w_i W(\mu, \nu^{(i)}).
\end{aligned}
\tag{29}
$$

$\qquad\square$

## B COMPARISON BETWEEN TOTAL VARIATION AND WASSERSTEIN DISTANCE

The total variation (TV) distance between two univariate distributions is defined as half of the absolute area between their probability density functions (PDFs). For instance, given two distributions $\mu$ and $\nu$ with PDFs $p_\mu$ and $p_\nu$, respectively, on space $\mathbb{R}_{\geq 0}$, the TV distance is given by

$$TV(\mu, \nu) = \frac{1}{2} \int_{\mathbb{R}_{\geq 0}} |p_\mu(x) - p_\nu(x)| \, dx \,. \tag{30}$$

In contrast, we expand $W(\mu, \nu)$ according to Eq. (28) by

$$
\begin{aligned}
W(\mu, \nu) &= \int_{\mathbb{R}_{\geq 0}} |F_\mu(x) - F_\nu(x)| \, dx = \int_{\mathbb{R}_{\geq 0}} \left| \int_0^x p_\mu(t) \, dt - \int_0^x p_\nu(t) \, dt \right| dx \\
&= \int_{\mathbb{R}_{\geq 0}} \left| \int_0^x p_\mu(t) - p_\nu(t) \, dt \right| dx \,.
\end{aligned}
\tag{31}
$$

The inner integration between 0 and $x$ indicates Wasserstein distance cares where two distributions $\mu$ and $\nu$ differ, whereas the total variation distance in Eq. (30) does not take this into consideration.

We would like to introduce a toy example to illustrate further why total variation distance can not consistently capture the closeness between two cumulative distribution functions (CDFs). Consider three conformal score distributions $P_V, Q_V^{(1)}, Q_V^{(2)}$ on space $\mathbb{R}_{\geq 0}$ with their PDFs:

$$p_{P_V}(v) = 1, v \in [0, 1];$$

$$p_{Q_V^{(1)}}(v) = \begin{cases} 1 & \text{if } v \in [0, 0.9], \\ 2 & \text{if } v \in (0.9, 0.95]; \end{cases}$$

$$p_{Q_V^{(2)}}(v) = \begin{cases} 2 & \text{if } v \in [0, 0.04], \\ 1 & \text{if } v \in (0.04, 0.96]. \end{cases}$$

Therefore, we calculate $TV(P_V, Q_V^{(1)}) = 0.05$ and $TV(P_V, Q_V^{(2)}) = 0.04$, while $W(P_V, Q_V^{(1)}) = 0.0025$ and $W(P_V, Q_V^{(2)}) = 0.0384$. In this example, a reduction in total variation distance results in a larger Wasserstein distance between two CDFs. Intuitively, TVD only measures the overall difference between two distributions without accounting for the specific locations where they diverge. In contrast, the Wasserstein distance will be high when divergence occurs early (i.e., at a small quantile), especially if the discrepancy persists until the "lagging" CDF catches up. We visualize the example in Figure 6.

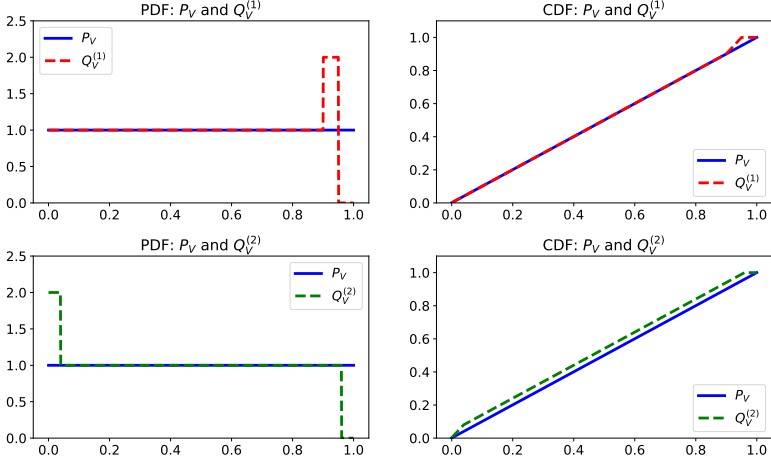

Figure 6: **Comparison between total variation distance and Wasserstein distance**: a reduction in the total variation distance does not necessarily result in CDFs becoming closer.

## C  RATIONALE FOR DIFFERENTIATING COVARIATE AND CONCEPT SHIFTS

There are two key reasons to differentiate between covariate and concept shifts. First, making this distinction enables the application of importance weighting. Minimizing the Wasserstein regularization term inevitably increases prediction residuals. By applying importance weighting, we expect to reduce the distance, mitigating the adverse effects of regularization on optimizing the regression loss function in Eq. (19). Figure 3 shows this expectation is met on five out of the six datasets. This occurs because, in most cases, covariate shifts exacerbate the distance caused by concept shifts ($f_P \neq f_Q$). Consequently, importance weighting effectively reduces this distance, as illustrated in Figure 7(a) and evidenced by the results for the airfoil self-noise, PeMSD4, PeMSD8, U.S.-States, and Japan-Prefectures datasets in Figure 3. However, there are instances where covariate shifts can alleviate the Wasserstein distance induced by concept shifts. In such cases, applying importance weighting may increase the distance, as demonstrated in the results for the Seattle-loop dataset in Figure 3. This phenomenon is further illustrated in Figure 7(b).

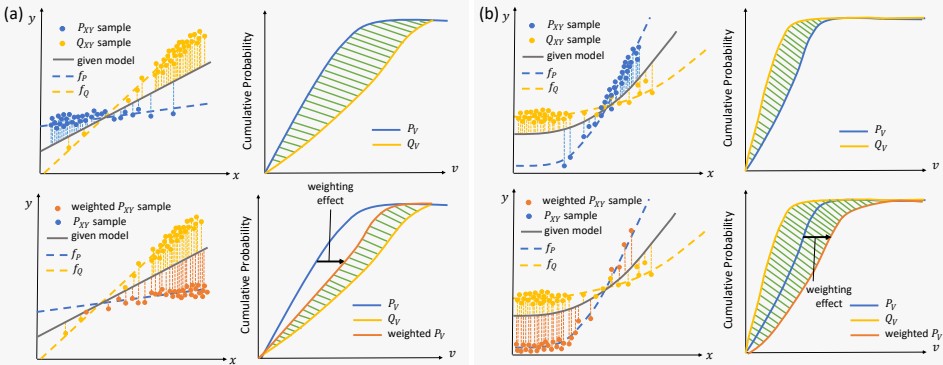

Figure 7: **Effect of importance weighting on Wasserstein distance:** (a) Scenario where importance weighting reduces Wasserstein distance; (b) Scenario where importance weighting enlarges Wasserstein distance.

Secondly, in multi-source CP, different training distributions $D_{XY}^{(i)}$ can suffer from different degrees of covariate and concept shifts. Importance weighting allows the regularized loss in Eq. (19) to minimize the distance between training conformal score distribution $D_V^{(i)}$ and its correspondingly weighted calibration conformal score distribution $D_{V,s_P}^{(i)}$, so the model can be more targeted on those whose remaining Wasserstein distances are large. Also, since various non-exchangeable test distributions will weight calibration conformal score distribution differently in the inference phase, prediction set sizes can be adaptive to different test distributions. In contrast, without importance weighting, the model can only regularize $\sum_{i=1}^{k} W(P_V, D_V^{(i)})$, and use the same quantile of $P_V$ to generate prediction sets for samples from all test distributions, resulting in the same prediction set size and lack of adaptiveness.

To further demonstrate the two reasons we mentioned above, we modify Wasserstein-regularization based on unweighted calibration conformal scores (i.e. $\sum_{i=1}^{k} W(P_V, D_V^{(i)})$) during training. Also, the weighting operation in the prediction phase in Algorithm 1 is removed accordingly. This method is denoted as WR-CP(uw). We performed WR-CP(uw) on the sampled data from the 10 trials of each dataset at $\alpha = 0.2$ and compared its results with those of WR-CP.

The comparison is depicted in Figure 8. Although the average coverage gaps between WR-CP and WR-CP(uw) are quite similar, at $3.1\%$ and $2.3\%$ respectively, the average prediction set size for WR-CP is $28.0\%$ smaller than that of WR-CP(uw). This observation proves our first reason that importance weighting effectively reduces the Wasserstein distance between calibration and test conformal scores. By doing so, it mitigates the side effect of optimizing the regularized objective function in Eq. (19), which increases prediction residuals. Since larger residuals result in larger prediction sets, reducing residuals directly helps minimize prediction set size. Additionally, the standard deviations of the prediction set sizes observed in WR-CP(uw) are typically smaller than those found in WR-CP. This proves the second reason that removing importance weighting will make prediction sets less adaptive to different test distributions.

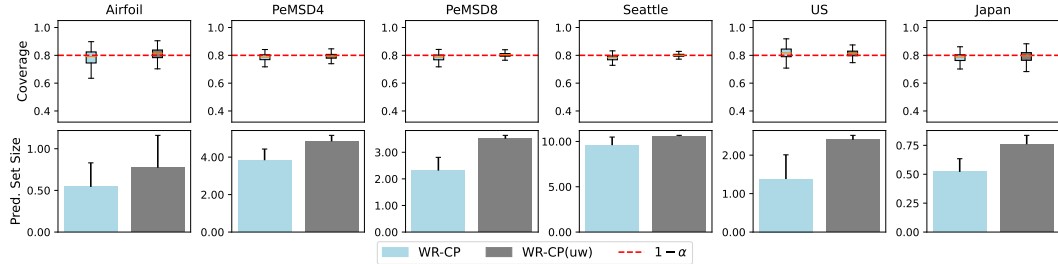

Figure 8: **Comparison between WR-CP and WR-CP(uw) at $\alpha = 0.2$.** Both methods were implemented using the same $\beta$ values of 4.5, 9, 9, 6, 8, and 20 across the datasets.

## D  GEOMETRIC INTUITION OF $\eta$

To provide a geometric intuition of $\eta$, we expand the definition of $\eta$ as

$$
\begin{aligned}
\eta &= \max_{x_1, x_2 \in \mathcal{X}} \frac{|s_P(x_1) - s_Q(x_2)|}{|f_P(x_1) - f_Q(x_2)|} \\
&= \max_{x_1, x_2 \in \mathcal{X}} \frac{|s(x_1, f_P(x_1)) - s(x_2, f_Q(x_2))|}{|f_P(x_1) - f_Q(x_2)|} \\
&= \max_{x_1, x_2 \in \mathcal{X}} \frac{||h(x_1) - f_P(x_1)| - |h(x_2) - f_Q(x_2)||}{|f_P(x_1) - f_Q(x_2)|}.
\end{aligned}
\tag{32}
$$

We first simplify the definition by assuming $x_1 = x_2$, so the denominator is the absolute difference between two ground-truth mapping functions $f_P$ and $f_Q$ at $x_1$, and the numerator is the absolute difference of the residuals of $f_P$ and $f_Q$ with a given model $h$ at $x_1$. $\eta$ is the largest ratio between the two absolute differences. A small $\eta$ means even if $f_P$ and $f_Q$ differ significantly, $h$ results in similar prediction residuals on $f_P$ and $f_Q$. When $x_1 \neq x_2$, $\eta$ is the largest ratio of the two absolute differences at two positions, $x_1$ and $x_2$, so a small $\eta$ means that $h$ can lead to similar residuals when $f_P(x_1)$ and $f_Q(x_2)$ differ. The expanded definition above includes both $x_1 = x_2$ and $x_1 \neq x_2$ conditions and Figure 9 (a) and (b) present the two conditions, respectively. Intuitively, the residual difference caused by concept shift will be constrained by $\eta$.

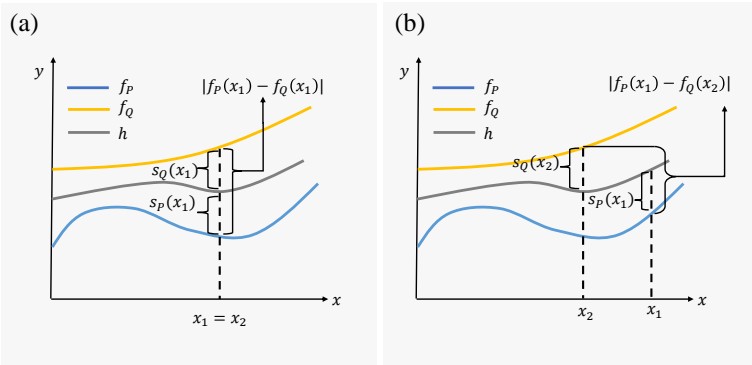

Figure 9: **Geometric intuition of $\eta$ when (a) $x_1 = x_2$ and (b) $x_1 \neq x_2$:** Intuitively, the residual difference caused by concept shift will be constrained by $\eta$.

# E    DISTRIBUTION ESTIMATION

## E.1    KERNEL DENSITY ESTIMATION

$\hat{P}_X$ and $\hat{D}_X^{(i)}$ for $i = 1, ..., k$ are obtained by kernel density estimation (KDE), and based on the estimated distributions we calculate the likelihood ratio.

In our experiments, we applied the Gaussian kernel, which is a positive function of $x \in \mathcal{X} \subseteq \mathbb{R}^d$ given by

$$\mathrm{K}(x, b) = \frac{1}{(\sqrt{2\pi}b)^d} e^{-\frac{\|x\|^2}{2b^2}}, \tag{33}$$

where $\|\cdot\|$ is Euclidean distance and $b$ is bandwidth. Given this kernel form, the estimated probability density, denoted by $\hat{p}$, at a position $x_a$ within a group of points $x_1, ..., x_n$ is

$$\hat{p}(x_a, \mathrm{K}) = \sum_{i=1}^{n} \mathrm{K}(x_a - x_i, b). \tag{34}$$

To find the optimized bandwidth value of $\hat{P}_X$ and $\hat{D}_X^{(i)}$ for $i = 1, ..., k$ on each dataset, we applied the grid search method with a bandwidth pool using scikit-learn package (Pedregosa et al., 2011). With the approximated marginal distribution densities, we can calculate the likelihood ratio to implement the weighting technique proposed by Tibshirani et al. (2019).

## E.2    POINT-WISE DISTRIBUTION ESTIMATION

$\hat{D}_V^{(i)}$ and $\hat{D}_{V, s_P}^{(i)}$ for $i = 1, ..., k$ are estimated as discontinuous, point-wise distributions to ensure differentiability during training. Specifically, as $\hat{D}_V^{(i)}$ and $\hat{D}_{V, s_P}^{(i)}$ are conformal score distributions on real number set $\mathbb{R}$, $W(\hat{D}_V^{(i)}, \hat{D}_{V, s_P}^{(i)})$ is equal to area between their CDFs, as Eq. (28) shows. Hence, our focus is on estimating the CDFs of $\hat{D}_V^{(i)}$ and $\hat{D}_{V, s_P}^{(i)}$ for $i = 1, ..., k$.

For the details of point-wise distribution estimation, consider we have a $x_1, ..., x_n$ drawn from a probability measure $\mu$ in space $\mathcal{X} \subseteq \mathbb{R}$, so the approximated CDF of $\mu$ is given by

$$F_{\hat{\mu}}(x) = \frac{1}{n} \sum_{j=1}^{n} \delta_{x_i} \mathbb{1}_{x_i < x}, \tag{35}$$

where $\mathbb{1}$ is the indicator function and $\delta_{x_i}$ represents the point mass at $x_i$ (i.e., the distribution placing all mass at the value $x_i$). In other words, Eq. (35) counts the partition of samples that are smaller than $x$. This point-wise estimation ensures that the Wasserstein-1 distance between the estimated distributions is differentiable.

# F    SUPPLEMENTARY EXPERIMENTAL INSIGHTS

## F.1    DATASETS

The airfoil self-noise dataset from the UCI Machine Learning Repository (Brooks & Marcolini, 2014) was intentionally modified to introduce covariate shift and concept shift among them. It includes 1503 instances. The target variable is the scaled sound pressure level of NASA airfoils, and there are 5 features: log frequency, angle of attack, chord length, free-stream velocity, and log displacement thickness of the suction side. To introduce covariate shift, we divided the original dataset into three subsets based on the 33% and 66% quantiles of the first dimension feature, log frequency, and partially shuffled them. Therefore, $k = 3$ for this dataset. We further introduced concept shifts among the three subsets by modifying target values. With $\xi$ following a normal distribution $N(0, 10)$, for $y$ in the first set, $y+ = y/1000 * \xi$; for $y$ in the second set, $y+ = y/\xi$; for $y$ in the third set, $y+ = \xi$. With the modified data, we conducted sampling trials to generate 10 randomly sampled datasets.

The Seattle-loop dataset Cui et al. (2019), as well as the PeMSD4 and PeMSED8 datasets Guo et al. (2019), consist of sensor-observed traffic volume and speed data gathered in Seattle, San Francisco, and San Bernardino, respectively. The data was collected at 5-minute intervals. Our goal for each dataset is to forecast the traffic speed of a specific interested local road segment in the next time step

by utilizing the current traffic speed and volume data from both the local segment and its neighboring segments. Before sampling, we selected 10 segments of interest for each dataset randomly, setting $k = 10$ for them. There are natural joint distribution shifts present among these segments because of the varying local traffic patterns.

The U.S.-States and Japan-Prefectures datasets Deng et al. (2020) contain data on the number of patients infected with influenza-like illness (ILI) reported by the U.S. Department of Health and Human Services, Center for Disease Control and Prevention (CDC), and the Japan Infectious Diseases Weekly Report, respectively. The data in each dataset is structured based on the collection region. Our objective is to utilize the regional predictive features, including population, the increase in the number of infected patients observed in the current week, and the annual cumulative total of infections, to forecast the rise in infections for the following week in the corresponding region. We also randomly selected 10 regions for both datasets, so $k = 10$. Due to the diverse regional epidemiological conditions, there are inherent joint distribution shifts among these regions.

For each dataset, we began by sampling $\mathcal{S}_{XY}^{(i)}$ from each subset $i$, for $i = 1, ..., k$, without replacement. After this step, we allocated the remaining elements within each subset for calibration and testing purposes. The parts intended for calibration across all subsets were then unified to form $\mathcal{S}_{XY}^{P}$. Lastly, to create diverse testing scenarios, we generated multiple test sets by randomly mixing the parts designated for testing from each subset with replacement. For each dataset, we conducted the sampling trial for 10 times, and calculated the mean and standard deviation of the results from these trials, as shown in Figure 3, Figure 4, and Figure 5. For efficiency, all CP methods were conducted as split conformal prediction.

We introduce a toy example to further illustrate that exchangeability does not hold. Consider we have two training distributions:

$$D_{XY}^{(1)} = N\left([0,0], \begin{bmatrix} 1 & 0.7 \\ 0.7 & 1 \end{bmatrix}\right); D_{XY}^{(2)} = N\left([1,1], \begin{bmatrix} 1 & -0.6 \\ -0.6 & 1 \end{bmatrix}\right).$$

A calibration distribution is a mixture of these two training distributions with known weights, such as a uniformly weighted mixture ($w_1 = w_2 = 0.5$). A test distribution is a mixture of $D_{XY}^{(1)}$ and $D_{XY}^{(2)}$ with unknown random weights. To visualize the non-exchangeability in Figure 10, we assume the unknown test distribution has weights of 0.2 for $D_{XY}^{(1)}$ and 0.8 for $D_{XY}^{(2)}$.

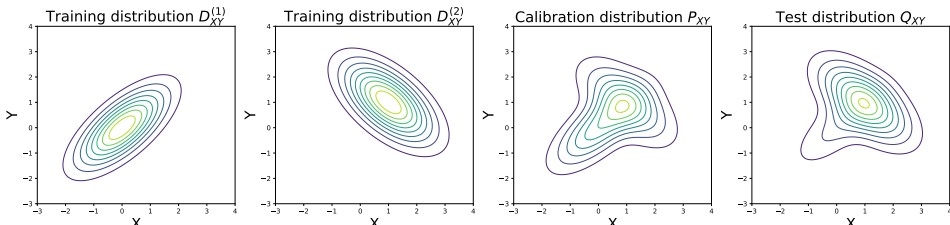

Figure 10: **Calibration and test samples are not exchangeable as they are from different distributions.**

### F.2 SPEARMAN'S COEFFICIENT

We first provide the definition of Pearson coefficient.

**Definition 5** (Pearson coefficient). *With $n$ pairs of samples, $(x_i, y_i)$ for $i = 1, ..., n$, of two random variables $X$ and $Y$, Pearson coefficient, $r_p$, is calculated as the covariance of the samples divided by the product of their standard deviations. Formally, it is given by*

$$r_p = \frac{\sum_{i=1}^{n}(x_i - \overline{x})(y_i - \overline{y})}{\sqrt{\sum_{i=1}^{n}(x_i - \overline{x})^2}\sqrt{\sum_{i=1}^{n}(y_i - \overline{y})^2}}, \tag{36}$$

*where $\overline{x}$ and $\overline{y}$ are the means of the samples of $X$ and $Y$, respectively.*

Based on Pearson coefficient, the definition of Spearman's coefficient is given as follows.

**Definition 6** (Spearman's coefficient). *With $n$ pairs of samples, $(x_i, y_i)$ for $i = 1, ..., n$, of two random variables $X$ and $Y$, letting $r(\cdot)$ be the rank function (i.e., $r(x_1) = 3$ indicates that $x_1$ is the third largest sample among $x_1, ..., x_n$), Spearman's coefficient, $r_s$, is defined as the Pearson coefficient between the ranked samples:*

$$r_s = \frac{\sum_{i=1}^{n} \left( r(x_i) - r(\overline{x}) \right) \left( r(y_i) - r(\overline{y}) \right)}{\sqrt{\sum_{i=1}^{n} \left( r(x_i) - r(\overline{x}) \right)^2} \sqrt{\sum_{i=1}^{n} \left( r(y_i) - r(\overline{y}) \right)^2}}, \tag{37}$$

*where $\overline{x}$ and $\overline{y}$ are the means of the samples of $X$ and $Y$, respectively.*

We calculated Spearman's coefficient between each distance measure and the largest coverage gap in Section 6 to confirm that Wasserstein distance holds the strongest positive correlation compared with other distance measures.

### F.3 Additional experiment results of subsection 6.4

In addition to the results shown in Figure 4, we present further experimental findings from Subsection 6.4 with $\alpha$ values of 0.1, 0.3, 0.4, 0.5, 0.6, 0.7, 0.8, and 0.9 on Figure 11, 12, 13, 14, 15, 16, 17, and 18, respectively. Clearly, WR-CP demonstrates the ability to generate more tightly concentrated coverages near $1 - \alpha$ compared to vanilla CP and IW-CP. Additionally, it yields smaller prediction set sizes than the state-of-the-art method WC-CP. These figures also reveal a trend where as the $\alpha$ value increases, WR-CP requires a smaller $\beta$ to achieve acceptable coverages around $1 - \alpha$, so the prediction set sizes produced by WR-CP are closer with those of vanilla CP and IW-CP, as evidenced by the results on the PeMSD4 in Figure 11 and Figure 18. This phenomenon could be attributed to the trade-off between conformal prediction accuracy and efficiency under joint distribution shift. The Wasserstein regularization term in Eq. (19) tends to prioritize aligning smaller conformal scores initially, as it reduces the Wasserstein penalty with a lesser increase in the empirical risk minimization term. Hence, as the hyperparameter $\beta$ increases, the model gradually aligns larger conformal scores from two different distributions, which will adversely impact the risk-driven term more. When considering a higher $\alpha$ value, the focus is on ensuring that the coverages on test data are close to the smaller $1 - \alpha$, indicating the importance of aligning small conformal scores. Consequently, a high $\beta$ value is not necessary in this case, leading to smaller prediction set sizes being achieved.

### F.4 Experiment setups in ablation study

To visualize a comprehensive and evenly-distributed set of optimal solutions on Pareto fronts, we utilized WR-CP with varying values of $\beta$ to produce the results depicted in Figure 5. As mentioned in Section 5, it is worth noting that when $\beta = 0$, WR-CP reverts to IW-CP. The selected $\beta$ values for the results of Figure 5 are shown in Table 2.

Table 2: $\beta$ values of WR-CP in ablation study

| Dataset | $\beta$ values |
|---------|----------------|
| Airfoil | 1, 1.5, 2, 2.5, 3, 3.5, 4.5, 6, 8, 9, 13, 20. |
| PeMSD4 | 1, 1.5, 2, 2.5, 3, 5, 7, 9, 11, 15, 20. |
| PeMSD8 | 1, 1.5, 2, 2.5, 3, 4, 5, 7, 9, 17. |
| Seattle | 1, 2, 3, 4, 4.5, 5, 5.5, 6, 7, 8, 10, 13, 15, 20. |
| U.S. | 1, 1.5, 2, 2.5, 3, 5, 6, 8, 13. |
| Japan | 1, 2, 3, 4, 6, 8, 10, 13, 20. |

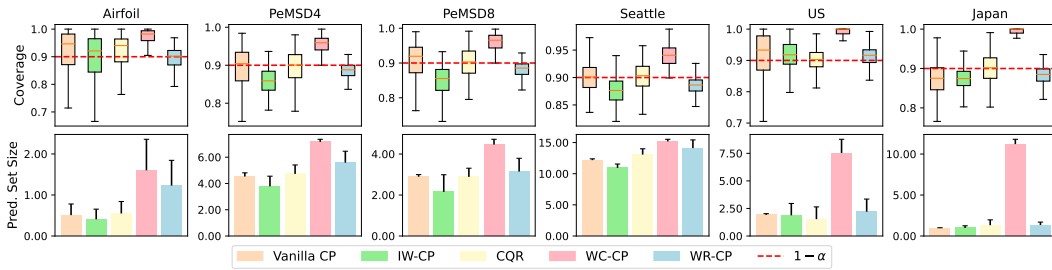

Figure 11: **Coverages and set sizes of WR-CP and baselines with $\alpha = 0.1$:** The $\beta$ values for the WR-CP method are 9, 11, 9, 8, 13, and 20, respectively.

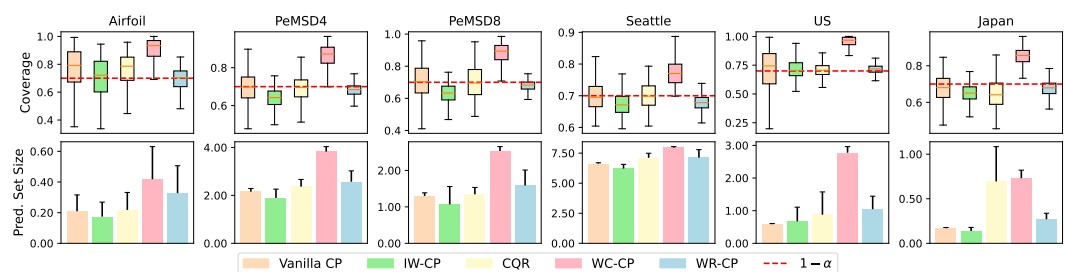

Figure 12: **Coverages and set sizes of WR-CP and baselines with $\alpha = 0.3$:** The $\beta$ values for the WR-CP method are 3, 5, 5, 5, 8, and 13, respectively.

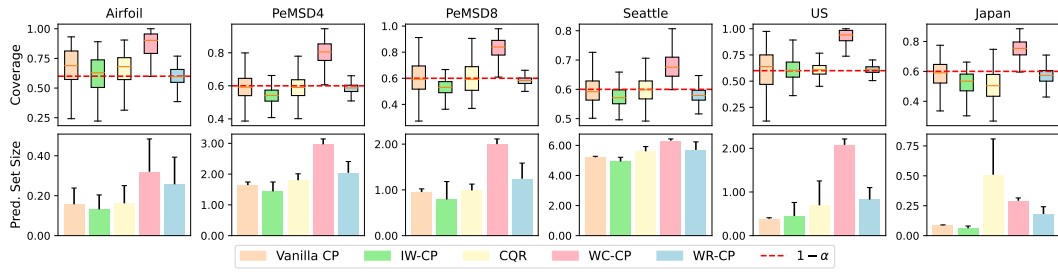

Figure 13: **Coverages and set sizes of WR-CP and baselines with $\alpha = 0.4$:** The $\beta$ values for the WR-CP method are 3, 5, 5, 5, 8, and 13, respectively.

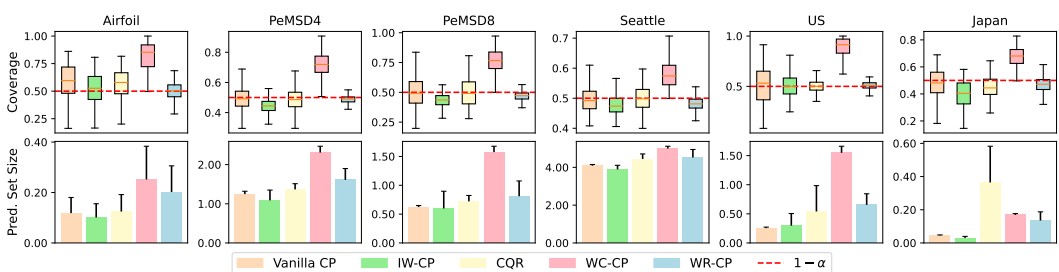

Figure 14: **Coverages and set sizes of WR-CP and baselines with $\alpha = 0.5$:** The $\beta$ values for the WR-CP method are 3, 5, 3, 5, 8, and 13, respectively.

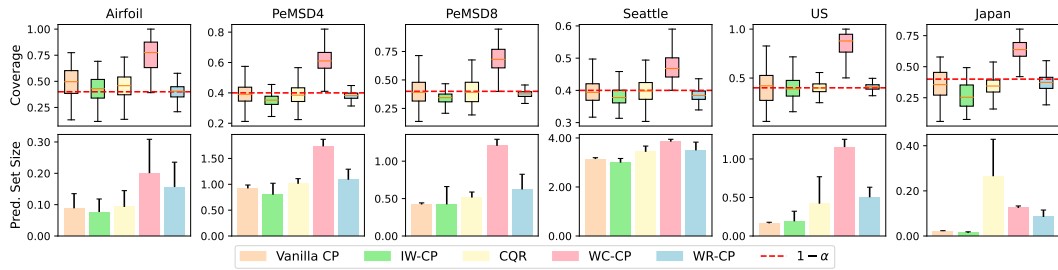

Figure 15: **Coverages and set sizes of WR-CP and baselines with** $\alpha = 0.6$**:** The $\beta$ values for the WR-CP method are 3, 5, 3, 5, 8, and 13, respectively.

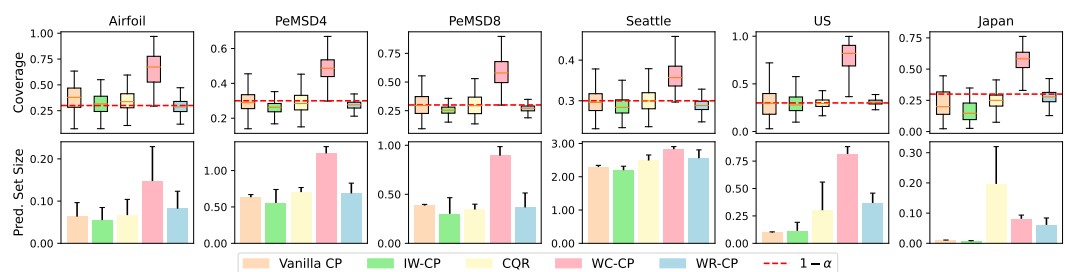

Figure 16: **Coverages and set sizes of WR-CP and baselines with** $\alpha = 0.7$**:** The $\beta$ values for the WR-CP method are 2, 2, 2, 5, 8, and 10, respectively.

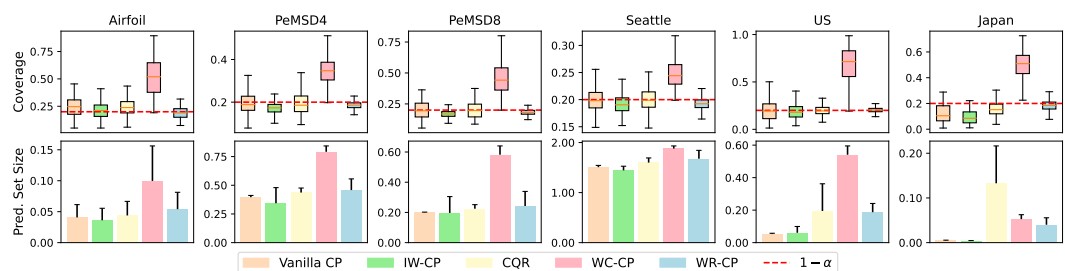

Figure 17: **Coverages and set sizes of WR-CP and baselines with** $\alpha = 0.8$**:** The $\beta$ values for the WR-CP method are 2, 2, 2, 5, 5, and 10, respectively.

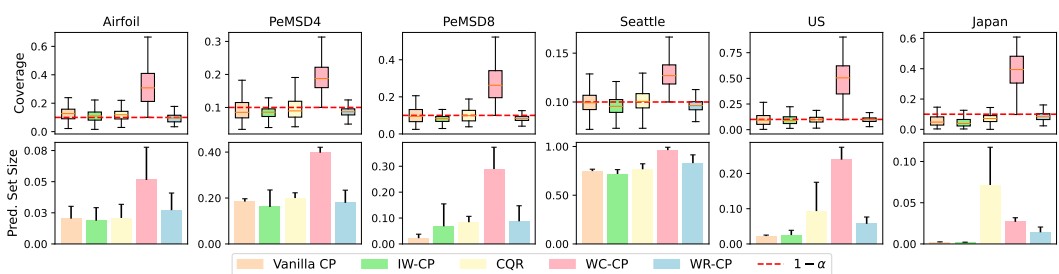

Figure 18: **Coverages and set sizes of WR-CP and baselines with** $\alpha = 0.9$**:** The $\beta$ values for the WR-CP method are 2, 1, 1, 5, 2, and 6, respectively.

## G PREDICTION EFFICIENCY WITH COVERAGE GUARANTEE

Although Wasserstein-regularized loss in Eq. (19) offers a controllable trade-off with significantly improved prediction efficiency and a mild coverage loss, it is worth investigating if this efficiency can be achieved with a coverage guarantee. In this section, we first derive a coverage lower bound of WR-CP via the multi-source setup in Appendix G.1. Then, we show that the combination of WC-CP and the Wasserstein-regularized loss can not achieve small prediction sets with ensured coverage in Appendix G.2.

### G.1 COVERAGE GUARANTEE FROM MULTI-SOURCE SETUP

Under the setup of multi-source conformal prediction, with $\tau$ as the $1 - \alpha$ quantile of the weighted calibration conformal score distribution $Q_{V,s_P}$, we can derive the coverage gap upper bound by

$$
\begin{aligned}
|F_{Q_{V,s_P}}(\tau) - F_{Q_V}(\tau)| &= \left| \sum_{i=1}^{k} w_i F_{D_{V,s_P}^{(i)}}(\tau) - \sum_{i=1}^{k} w_i F_{D_V^{(i)}}(\tau) \right| \\
&\leq \sum_{i=1}^{k} w_i |F_{D_{V,s_P}^{(i)}}(\tau) - F_{D_V^{(i)}}(\tau)| \\
&\leq \sup_{i \in \{1,\ldots,k\}} |F_{D_{V,s_P}^{(i)}}(\tau) - F_{D_V^{(i)}}(\tau)|.
\end{aligned}
\tag{38}
$$

In other words, the coverage gap on a test distribution must be less or equal to the largest gap at $\tau$ among multiple training distributions. Denoting $\alpha_D = \sup_{i \in \{1,\ldots,k\}} |F_{D_{V,s_P}^{(i)}}(\tau) - F_{D_V^{(i)}}(\tau)|$, we have a coverage guarantee $\Pr(Y_{n+1} \in X_{n+1}) \geq 1 - \alpha - \alpha_D$.

The regularization term $\sum_{i=1}^{k} W(D_{V,s_P}^{(i)}, D_V^{(i)})$ in Eq. (19) can minimize $\alpha_D$, and thus making $1 - \alpha - \alpha_D$ closer to the desired $1 - \alpha$. It is important to highlight that $\alpha_D$ is adaptive to variations in test distribution $Q_V$, as evident from Eq. (38). This adaptivity ensures that the lower bound dynamically adjusts to different $Q_V$. To evaluate the prediction efficiency of WR-CP under this guarantee, we set $\alpha = 0.1$ and computed the corresponding $\alpha_D$ for various test distributions. Additionally, we calculated the coverage and prediction set size of WC-CP on each test distribution, using the corresponding guarantee at $1 - \alpha - \alpha_D$ for comparison.

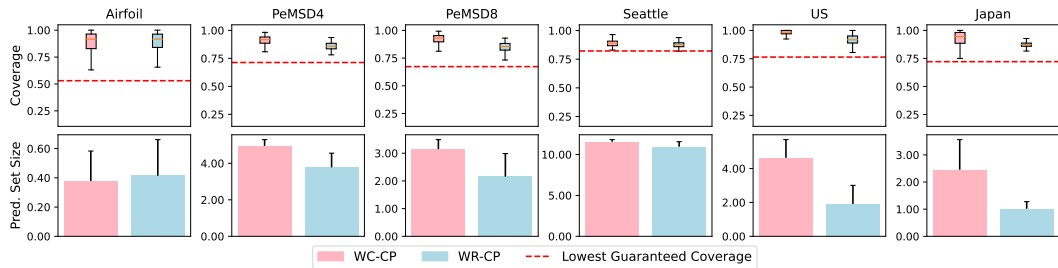

Figure 19: **Coverages and set sizes of WC-CP and WR-CP with coverage guarantee at** $1 - \alpha - \alpha_D$**.**

The experiment results are depicted in Figure 19, demonstrating improved prediction efficiency on the PeMSD4, PeMSD8, U.S.-States, and Japan-Prefectures datasets. However, the efficiency remains almost unchanged on the Seattle-loop dataset and even declines on the airfoil self-noise dataset. This phenomenon can be attributed to the regularization mechanism. While WR-CP enhances prediction efficiency by leveraging the calibration distribution to generate prediction sets, regularization inevitably increases prediction residuals, leading to larger prediction sets. These two opposing effects can interact differently depending on the dataset characteristics. When the efficiency gains outweigh the drawbacks of regularization, we observe reduced prediction set size. Conversely, in datasets like the Seattle-loop and airfoil self-noise, the benefits of regularization are outweighed by the increased prediction residuals, resulting in unchanged or diminished efficiency. The averaged prediction set size reduction across the six datasets is $26.9\%$.

## G.2 POOR COMPATIBILITY BETWEEN WASSERSTEIN-REGULARIZED LOSS AND WC-CP

Since the WC-CP is a conservative *post-hoc* uncertainty quantification method but the proposed regularized loss in Eq. (19) is applied during *training*, one may consider applying WC-CP upon the model trained by the regularized loss to obtain guaranteed coverage. However, WC-CP and the model are not suitable for complementing each other. While regularization enhances the reliability of calibration distributions, the worst-case approach depends exclusively on the upper bound of $1 - \alpha$ test conformal score quantile, rendering it unable to benefit from regularization. In contrast, the WC-CP may result in larger prediction sets under this condition, as the regularization inevitably increases the prediction residuals, which in turn increases the upper bound of the test conformal score quantile. Experiment results in Figure 20 demonstrate the analysis, where WC-CP is the worst-case method based on a residual-driven model (same as the WC-CP method in Section 6.4), and Hybrid WC-WR represents applying WC-CP to a model trained by Eq. (19).

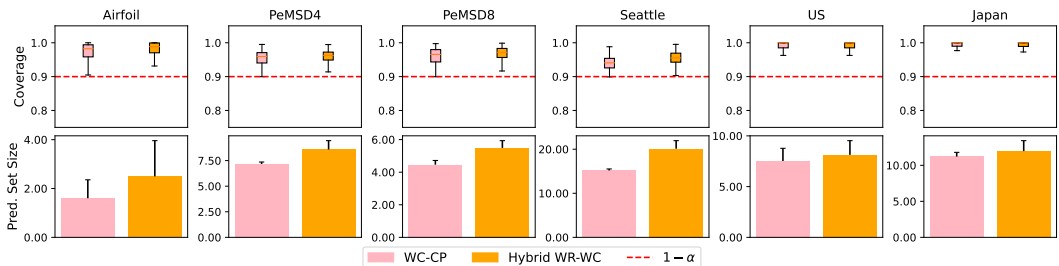

Figure 20: **Coverages and set sizes of WC-CP and Hybrid WC-WR with coverage guarantee** $1 - \alpha = 0.9$.

# H LIMITATIONS

## H.1 SUSCEPTIBILITY TO DENSITY ESTIMATION ERRORS

Given that Wasserstein regularization relies on importance-weighted conformal scores, its performance is greatly influenced by the accuracy of the estimated likelihood ratio obtained through KDE. Inaccurate estimation can significantly impact the effectiveness of WR-CP. For instance, in Figure 4, WR-CP yields larger prediction set sizes with less concentrated coverages on the airfoil self-noise dataset compared to other datasets. This can be attributed to the airfoil self-noise dataset having the highest feature dimension (5) and the smallest size of the sampled $\mathcal{S}_{XY}^{P}$ (500). These challenges in KDE lead to suboptimal performance of WR-CP on the airfoil self-noise dataset when compared to its performance on others.

The main reason for KDE error is numerical instability, which can arise from several factors. A poor choice of kernel is a critical contributor; for instance, kernels with sharp edges or discontinuities, such as rectangular or triangular kernels, can result in jagged density estimates and amplify errors near boundaries. Fat-tailed kernels, such as the Cauchy kernel, may assign excessive weight to distant data points, leading to inaccuracies in density estimates and numerical precision challenges. Additionally, the lack of feature normalization can exacerbate the effects of extreme values, skewing the density estimation process and reducing computational stability. Lastly, inappropriate bandwidth selection, either too small (overfitting) or too large (underfitting), can disrupt the balance between bias and variance, further contributing to instability in the estimation.

In our work, we first adopted the Gaussian kernel, valued for its smoothness and numerical stability. To mitigate the influence of extreme values, we applied feature normalization, ensuring a more stable density estimation process. Additionally, we conducted a comprehensive grid search to fine-tune the bandwidth, achieving an optimal balance between bias and variance for robust and accurate results. The bandwidth candidates were selected from a logarithmically spaced range between $10^{-2}$ and $10^{0.5}$, consisting of 20 evenly distributed values on a logarithmic scale.

## H.2 COMPUTATIONAL CHALLENGES IN KDE

We applied a grid search approach to identify the optimal bandwidth for KDE, which ensures an effective balance between bias and variance in density estimation. However, this method often involves extensive computational effort, particularly when working with high-dimensional datasets, as it requires repeated calculations over a range of bandwidth values. To address this challenge, Bernacchia–Pigolotti KDE (Bernacchia & Pigolotti, 2011) introduces an innovative framework that combines a Fourier-based filter with a systematic approach for simultaneously determining both the kernel shape and bandwidth. This method not only reduces subjectivity in kernel selection but also offers a more efficient computational pathway. Building on this foundation, FastKDE (O'Brien et al., 2016) adapts and extends the Bernacchia–Pigolotti approach for high-dimensional scenarios, incorporating optimizations that significantly improve computational speed and scalability. These advancements represent promising directions for mitigating the computational overhead in our own work, where similar strategies could be leveraged to streamline the bandwidth selection process and enhance the overall efficiency of KDE in complex datasets.

## H.3 OTHER CHOICES OF THE CALIBRATION DISTRIBUTION

In the experiments conducted in Section 6, we specifically examine the scenario where the calibration data follows a mixture distribution of $D_{XY}^{(i)}$ for $i = 1, ..., k$ with equal weights. However, this may not always be the case in real-world situations. Given that the calibration distribution plays a crucial role in determining the difficulty of minimizing Eq. (19) during training, it is valuable to investigate the performance of WR-CP with a calibration distribution different from a mixture of training distributions.

