# OpenReview forum: "Wasserstein-Regularized Conformal Prediction under General Distribution Shift"
_ICLR.cc/2025/Conference — ICLR 2025 Poster_

### Official Review · Reviewer_HBTg · 2024-10-29

**Soundness:** 3
**Presentation:** 2
**Contribution:** 2
**Rating:** 6
**Confidence:** 3

**Summary:**

Conformal Prediction (CP) sets may not have the guaranteed lower-bounded coverage if calibration and test samples are not exchangeable. The authors derive bounds for and optimize the gap between nominal and empirical coverage in such a distribution-shift setup. Unlike existing work, the bounds and the optimization loss are given in terms of the Wasserstein Distance.

**Strengths:**

- Improving CP under distribution shifts is timely and challenging. The provided bounds and algorithm may increase CP applicability to new kinds of data.
- Optimizing the uncertainty estimation of a CP algorithm is an increasingly popular topic. It is interesting to focus on reducing a model's uncertainty instead of improving the accuracy of its point predictions.
- Looking at CP-aware optimization as the joint optimization of a standard objective and a CP-aware regularization term.
- The proposed penalty term does not seem to involve the (non-differentiable) sample quantiles of the conformity score distribution.

**Weaknesses:**

- Using WD to optimize the prediction intervals is justified by saying that the Total Variation Distance (TVD) *ignores where two conformal score distributions differ* by focusing on the worst-case discrepancy and is location independent. The authors should clarify the difference between the integral and the infimum in the WD definition, Eq.5, and the sup appearing in the definition of TVD.
- The authors may justify better their idea of separating covariate and concept shift. In practice, such separation requires making strong assumptions on the data-generating distribution. The advantages are less clear.
- The optimization scheme assumes the availability of samples from the base and the shifted distributions. This is not the case in the standard distribution-shift setup. Access to the test samples at training time would allow a more straightforward approach where the CP algorithm is trained on all samples and hence not affected by distribution shift.
- The authors should discuss the exchangeability of the samples in the proposed application of Section 5.

**Questions:**

- If the goal is to optimize the size of the prediction intervals, why is it important to distinguish between covariate and concept shift? And why should one avoid the worst-case approach of an analogous TVD minimization?
- Why does the TVD fail *to indicate coverage gap changes thoroughly" and  *is agnostic about where two distributions diverge*? The WD integrates over local divergences and does not explicitly depend on $\alpha$. Can one prove that the TVD fails to summarize the distance between the quantiles of two distributions?
- Are the test and calibration samples of Section 5 non-exchangeable?
- Is the loss in Eq.19 tractable in practice? Doesn't the KDE cause numerical instabilities for bad kernel choices?
- Have you compared the proposed optimization with a naive CP baseline trained on all available original and shifted samples?
- Do the proofs of Theorems 1 and 2 depend on the specific conformity measure?

---

> ### Author Response · Authors · 2024-11-20
> **Responses to weakness 1,2 and question 1.**
>
> Thank you for your insightful comments and questions! We appreciate your feedback and would like to provide further clarification on our work. We consolidated several comments regarding the weaknesses and questions into a single response and addressed them collectively. References are listed at the end of the series of responses. Please note that we have uploaded a revised version, which includes additional details for your review.
>
> >The authors should clarify the difference between the integral and the infimum in the WD definition, Eq.5, and the sup appearing in the definition of TVD.
>
> The Wasserstein distance (W-distance) measures the cost of optimally transporting mass between two probability distributions. W-distance is also known as earth mover's distance. In Eq. (5), the integral represents the "cost" of transporting mass under a specific coupling $\gamma$, which is a joint distribution with marginals matching the two given distributions $\mu$ and $\nu$. The cost function is integrated over the product space. The infimum is taken over all possible couplings $\gamma$ that align the marginal distributions. It ensures that the W-distance represents the minimal transportation cost.
>
> The total variation distance is defined as the largest possible difference in probabilities assigned to the same event by the two distributions. For two distributions $\mu$ and $\nu$ on space $\mathcal{X}$, this is formalized as
> \begin{equation}
> TV(\mu,\nu)=\sup_{\mathcal{A} \subset \mathcal{X}}|\mu(\mathcal{A})-\nu(\mathcal{A})|.
> \end{equation}
> The supremum is over all measurable sets $\mathcal{A}$ and ensures that the distance captures the worst-case discrepancy between the two distributions. For probability density functions $p_\mu$ and $p_\nu$, this can be equivalently written as:
> \begin{equation}
> TV(\mu,\nu)=\frac{1}{2}\int_{\mathcal{X}}|p _\mu(x)-p _\nu(x)|.
> \end{equation}
>
> >In practice, such separation requires making strong assumptions on the data-generating distribution.
>
> Indeed, the separation is achieved by importance weighting, which requires us to have prior knowledge about test feature distribution $Q_X$, such as a batch of test inputs, thereby KDE and importance weighting are implementable. In other words, WR-CP can not work when test inputs are provided one by one, such as time-series data. We would like to highlight that this is a common consideration when implementing importance weighting, as seen in approaches like CoDrug[2], SLCP[3], and DP-FedCP[4], rather than a unique limitation of our work.
>
> > If the goal is to optimize the size of the prediction intervals, why is it important to distinguish between covariate and concept shift?
>
> There are two reasons for distinguishing covariate and concept shifts.
>
> First, the distinguishment allows the application of importance weighting. Minimizing the Wasserstein-regularization term inevitably increases prediction residuals. We expect importance weighting can reduce the distance, thus alleviating the side-effect of regularization on the optimization of the regression loss function. Figure 3 shows this expectation is met on five out of the six datasets. We also explained and visualized when importance weighting may not lead to a reduced W-distance in **Appendix D.3**. and **Figure 9** in the latest version.
>
> Secondly, in multi-source CP, different subpopulations $D _{XY}^{(i)}$ can suffer from different degrees of covariate and concept shifts. Importance weighting allows the regularized loss in Eq. (19) to minimize the distance between training conformal score distribution $D _{V}^{(i)}$ and its correspondingly weighted calibration conformal score distribution $D _{V, s _P}^{(i)}$,  so the model can be more targeted on those whose remaining W-distances are large. Also, since various non-exchangeable test distributions will weight calibration conformal score distribution differently in the inference phase, prediction set sizes can be adaptive to different test distributions. In contrast, without importance weighting, the model can only regularize $\sum _{i=1}^k W(P _V, D _V^{(i)})$, and use the same quantile of $P_V$ to generate prediction sets for samples from all test distributions, resulting in the same prediction set size and lack of adaptiveness.
>
> >And why should one avoid the worst-case approach of an analogous TVD minimization?
>
> It is important to note that total variation distance does not consistently capture the overall closeness between two cumulative distribution functions (CDFs). This point is further explained in the subsequent answer. As a result, focusing on minimizing total variation distance could potentially increase the W-distance between the CDFs. This outcome is undesirable when the goal is to achieve a small coverage gap across different levels of $\alpha$. **Figure 11-18** also experimentally demonstrated minimizing W-distance can achieve a good balance between coverage gap and prediction efficiency along different $\alpha$.

---

> ### Author Response · Authors · 2024-11-20
> **Responses to weakness 3 and question 2,3,4**
>
> >Why does the TVD fail *to indicate coverage gap changes thoroughly" and is agnostic about where two distributions diverge? The WD integrates over local divergences and does not explicitly depend on α. Can one prove that the TVD fails to summarize the distance between the quantiles of two distributions?
>
> The total variation distance of two distributions is defined as half of the absolute area between their probability density functions (PDFs). To illustrate why total variation distance can not capture the distance between the CDFs of two distributions, let us introduce a toy example. Consider three distributions $P_V$, $Q_V^{(1)}$, and $Q_V^{(2)}$ on space $\mathbb{R}$ with PDFs as
>
> \begin{equation}
>     p_{P _V^{ }}(v)=1,v\in[0,1];
> \end{equation}
>
> \begin{equation}
>     p _{Q _V^{(1)}}(v)= 1, \text{if $v\in[0,0.9]$}; p _{Q _V^{(1)}}(v)= 2, \text{if $v\in(0.9,0.95]$};
> \end{equation}
>
> \begin{equation}
>     p _{Q _V^{(2)}}(v)=2, \text{ if $v\in[0,0.04]$};   p _{Q _V^{(2)}}(v)=1, \text{ if $v\in(0.04,0.96]$}.
> \end{equation}
>
> Based on the definition of total variation distance, we have $TV(P _V, Q _V^{(1)})=0.05$, $TV(P _V, Q _V^{(2)})=0.04$. However, $Q _V^{(1)}$ is closer to $P_V$ than $Q _V^{(2)}$ in terms of CDFs, with $W(P_V,Q_V^{(1)})=0.0025$ and $W(P_V,Q_V^{(2)})=0.0384$. This example demonstrates that a reduced total variation distance does not necessarily result in CDFs becoming closer. Intuitively, total variation distance only measures the overall difference between two distributions without accounting for the specific locations where they diverge. In contrast, the W-distance will be high when divergence occurs early (i.e., at small quantiles), especially if the discrepancy persists until the "lagging" CDF catches up. We refer to the updated **Appendix F.1** and **Figure 18** for a plot of this example.
>
> > The optimization scheme assumes the availability of samples from the base and the shifted distributions. This is not the case in the standard distribution-shift setup. Access to the test samples at training time would allow a more straightforward approach where the CP algorithm is trained on all samples and hence not affected by distribution shift.
>
> First, during the preprocessing of the dataset to create training, calibration, and test samples, the sampling was performed without replacement, ensuring that each sample appeared in only one of the three sets, so we did not have access to the test samples during training. The detailed sampling procedure is introduced in **Appendix D.1**.
>
> Also, the training, calibration, and test samples represent different underlying distributions, so the exchangeability does not hold, as we introduced in Eq. (17).
>
> >Are the test and calibration samples of Section 5 non-exchangeable?
>
> The test and calibration samples in Section 5 are not exchangeable. The calibration samples are drawn from a known fixed distribution. For a test distribution $Q_{XY}$, its weight  $w_i$ of $D_{XY}^{(i)}$ for $i=1,...,k$ are random and unknown. so exchangeability does not hold. We introduce a toy example in **Figure 8** to demonstrate the non-exchangeability between calibration and test samples. In Section 6, we conduct experiments using multiple random test distributions with different weights to demonstrate the effectiveness of our approach. The number of test distributions ranges from 30 to 100 depending on the size of datasets.
>
> >Is the loss in Eq.19 tractable in practice?
>
> The loss function presented in Eq. (19) is both differentiable and tractable. Its regularization term represents the W-distance between two sets of conformal scores, as discussed in **Appendix C.2**. Alternatively, the Sinkhorn divergence can be used to approximate the W-distance. This approach can be readily implemented using the Geomloss package [1], although it was not utilized in this study.
>
> >Doesn't the KDE cause numerical instabilities for bad kernel choices?
>
> Yes, numerical instabilities in KDE can be caused by a poor choice of kernel. For instance, kernels with sharp edges or discontinuities, such as rectangular or triangular kernels, can result in jagged density estimates and amplify errors near boundaries. Similarly, fat-tailed kernels, such as the Cauchy kernel, may assign excessive weight to distant data points, leading to inaccurate density estimates and numerical precision issues.
>
> In our work, we utilized the Gaussian kernel for its smoothness, differentiability, and numerical stability. Additionally, we applied feature normalization to mitigate the effects of extreme values and improve computational precision. To ensure robust density estimation, we performed a grid search for bandwidth selection, achieving an optimal balance between bias and variance. Please check **Appendix C.1** for more details about KDE.

---

> > ### Comment · Reviewer_HBTg · 2024-11-28
> > **Thank you**
> >
> > I appreciate the authors' rebuttal and will rise my score to 6.
> >
> > I am still confused about the exchangeability in Section 5. As the weights are sampled uniformly and Q contains all training mixture components, isn t sampling from Q equivalent to sampling from the training set?

---

> ### Author Response · Authors · 2024-11-20
> **Responses to question 5,6**
>
> >Have you compared the proposed optimization with a naive CP baseline trained on all available original and shifted samples?
>
> We conducted the vanilla CP (naive CP) upon a residual-driven (i.e. unregularized) model trained with the samples from training subpopulations $D_{XY}^{(i)}$ for $i=1,...k$. Shifted test distributions should not be observed during training.
>
> >Do the proofs of Theorems 1 and 2 depend on the specific conformity measure?
>
> Theorems 1 and 2 are generalizable statistic analyses and study the properties of W-distance between two distributions when one is a pushforward measure of another, so these two theorems hold as far as calibration and test distributions satisfy Definition 3.
>
> **Reference**
>
> [1] Feydy, Jean, et al. "Interpolating between optimal transport and mmd using sinkhorn divergences." The 22nd International Conference on Artificial Intelligence and Statistics. PMLR, 2019.
>
> [2] Laghuvarapu, Siddhartha, Zhen Lin, and Jimeng Sun. "CoDrug: Conformal Drug Property Prediction with Density Estimation under Covariate Shift." Advances in Neural Information Processing Systems 36 (2024).
>
> [3] Han, Xing, et al. "Split localized conformal prediction." arXiv preprint arXiv:2206.13092 (2022).
>
> [4] Plassier, Vincent, et al. "Conformal prediction for federated uncertainty quantification under label shift." International Conference on Machine Learning. PMLR, 2023.

---

> > ### Author Response · Authors · 2024-11-22
> > **We appreciate your feedback and advices.**
> >
> > Thank you so much for your insightful feedback on this work. We hope that our responses and the additional clarifications provided in the appendix address your concerns. We would greatly appreciate any further thoughts or suggestions you may have to help us improve and refine the work further.

---

> ### Author Response · Authors · 2024-11-23
> **Additional Experiments Addressing the First Sub-Question of Question 1**
>
> >If the goal is to optimize the size of the prediction intervals, why is it important to distinguish between covariate and concept shift?
>
> To further illustrate the two reasons mentioned above in response to the sub-question, we modify Wasserstein-regularization by making it regularize via unweighted calibration conformal scores (i.e. $\sum _{i=1}^k W(P _V, D _V^{(i)})$) during training. Also, the weighting operation in the prediction phase in Algorithm 1 is removed accordingly. This method is denoted as WR-CP(uw). We performed WR-CP(uw) on the sampled data from the first trial of each dataset at $\alpha=0.2$, and compared its results with those of WR-CP. The average prediction set sizes of the two methods are listed below, with standard deviations shown in parentheses.
>
> |           |   Airfoil   |    PeMSD4   |    PeMSD8   |    Seattle   |     U.S.    |    Japan    |
> |:---------:|:-----------:|:-----------:|:-----------:|:------------:|:-----------:|:-----------:|
> |   WR-CP   | 0.27 (0.02) | 3.84 (0.44) | 2.49 (0.47) | 10.47 (1.12) | 1.63 (0.69) | 0.52 (0.10) |
> | WR-CP(uw) | 0.38 (0.00) | 4.98 (0.00) | 3.17 (0.00) | 10.63 (0.00) | 2.43 (0.00) | 0.71 (0.00) |
>
> The average coverage gaps between WR-CP and WR-CP(uw) are quite similar, at 2.7\% and 2.2\% respectively. However, as the table above shows, the average prediction set size for WR-CP is **22.3% smaller** than that of WR-CP(uw). This observation proves our first reason that importance weighting effectively reduces the Wasserstein distance between calibration and test conformal scores. By doing so, it mitigates the side effect of optimizing the regularized objective function in Eq.(19), which increases prediction residuals. Since larger residuals result in larger prediction sets, reducing residuals directly helps minimize prediction set size. The zero standard deviation observed for WR-CP(uw) proves the second reason that removing importance weighting will make prediction sets lack adaptiveness to different test distributions. Besides, we can observe that WR-CP does not perform better than WR-CP(uw) on the Seattle-loop dataset, reflecting that importance weighting does not reduce Wasserstein distance on that dataset in Figure 3. We refer to **Appendix D.3** and **Figure 10** for more detailed information about the experiment.

---

> > ### Author Response · Authors · 2024-11-24
> > **The discussion of KDE numerical instability is included in Appendix E.1.**
> >
> > >Doesn't the KDE cause numerical instabilities for bad kernel choices?
> >
> > We expanded **Appendix E.1** to address your concerns regarding numerical instability in KDE. In addition to discussing the impact of a poor kernel choice, we have included strategies to mitigate issues arising from extreme values and improper bandwidth selection. These measures are aimed at improving the accuracy and robustness of KDE, ensuring reliable density estimation even in challenging scenarios.  Moreover, we also added **Appendix E.2** to discuss how to improve the computation efficiency of KDE.

---

> ### Author Response · Authors · 2024-11-26
> **Additional analysis on the comparison between total variation distance and Wasserstein distance**
>
> We provide further analysis in **Appendix F.1** to mathematically demonstrate why the Wasserstein distance takes into account the specific locations where two distributions differ, whereas the total variation distance fails to consider this aspect. Please check the latest revised PDF.
>
> Besides, as the final day for authors to upload modified PDFs is approaching, we would greatly appreciate any feedback you may have on our rebuttal and work. We look forward to your thoughts.

---

> ### Author Response · Authors · 2024-11-29
> **Thank You for Your Constructive Review and Feedback**
>
> Thank you for your constructive feedback and thoughtful review of our submission. We greatly appreciate the time and effort you took to evaluate my work! We would like to provide further clarification regarding the non-exchangeability in the problem setup.
>
> The statement "sampling from $Q$ is equivalent to sampling from the training set" may not accurately describe our data sampling process. Specifically, we sampled the training sets $S_{XY}^{(i)}$ for $i = 1, \dots, k$, the calibration set $S_{XY}^{P}$, and the test set $S_{XY}^{Q}$ independently from the raw data without replacement. As a result, the test set $S_{XY}^{Q}$ is not a subset of either the training or calibration sets. As previously mentioned, each sample in the raw data appears in only one of the three sets.
>
> We believe you intended to convey that "sampling from the test distribution $Q_{XY}$ is sampling from the training distributions $D_{XY}^{(i)}$, since $Q_{XY}$ is a mixture of $D_{XY}^{(i)}, i=1,...,k$". This understanding is correct.
>
> However, although both calibration distribution $P_{XY}$ and test distribution $Q_{XY}$ are mixtures of $D_{XY}^{(i)}, i=1,...,k$, exchangeability does not hold (i.e. $P_{XY}\neq Q_{XY}$). The key distinction is that the calibration distribution $P_{XY}$ is a specific, known distribution—such as a uniformly weighted mixture—while **the weights of $D_{XY}^{(i)}$ in a test distribution $Q_{XY}$ are random and unknown (not uniform)**.  As a result, $P_{XY}\neq Q_{XY}$, and therefore calibration and test samples are non-exchangeable. This is further illustrated in **Figure 8**, where the test distribution (in the fourth subplot) differs from the calibration distribution (in the third subplot).
>
> We hope this clarification is helpful and thank you again for your insightful review.

---

### Official Review · Reviewer_GF9E · 2024-11-04

**Soundness:** 2
**Presentation:** 2
**Contribution:** 2
**Rating:** 6
**Confidence:** 3

**Summary:**

This paper investigates the effect of distribution shift between calibration and test data on conformal prediction guarantees. Theoretically, they upper bound the total distribution shift by two terms that decouple the effect of covariate and concept shifts. They are able to do so by using Wasserstein distance based upper bound of coverage gap ( which they also introduce.) Furthermore, they propose methods by which they minimize each term in the upper bound. Empirically they show that their proposed algorithm (WR-CP) can reduce coverage gaps and output smaller prediction sets.

**Strengths:**

1. They address an important problem in CP, namely decoupling the effect of covariate shift and concept shift on the coverage gap.
2. They upper bound the coverage gap by adopting the Wasserstein distance over the space of probability distribution of conformal scores. This allows to indicate changes across different values of mis-coverage rate $\alpha$.
3. The upper bound is then decomposed to explicitly depend on two terms where the effect of covariate/concept shift is decoupled. This is an important and novel contribution. The theory seems to extensively validate the results.

**Weaknesses:**

1. The minimization of the term corresponding to covariate shift, seems to just use previous contributions in the field (Tibshirani et. al 2019). Furthermore the optimization algorithm heavily depends on good density estimation, which is very difficult to do in practice.

2. The insights from the newly derived Wasserstein-based upper bound on the coverage-gap are the basis of the optimization procedure introduced in algorithm 1 (WR-CP). To this end:

2.a) There are no formal robustness/coverage guarantees on the final prediction sets. The way the problem is introduced, this becomes a major limitation.

2.b) The problem is considered within the context of "regression" tasks. However, when exploring the literature including classification tasks, we can see a clear distinction between training phase (where the predictor model parameters are changed in order to optimize a conformal-based loss ( e.g (Stutz et. al 2022, Yan et. al 2024)) , and the calibration procedure where the model is treated as a black-box and formal coverage guarantees are the main point. The same pipeline seamlessly extends to the regression setting as well. I do not believe the authors have made a clear distinction where their contribution lies in this context.

2.c) The experiments compare the proposed algorithm WR-CP to Worst-Case-CP ( WC-CP) to indicate a reduction in length-inefficiency of the prediction sets. I do not believe the experiments are a fair comparison, as the model parameters are "changed". The fair comparison would apply a fixed calibration procedure, with guarantees to both models ( one baseline trained without any insights from the Wasserstein upper bound, one trained according to WR-CP) in order to measure the gains in terms of coverage gap or the length-inefficiency. To indicate this point further, (https://lilywenglab.github.io/Provably-Robust-Conformal-Prediction/) applied "Conformal Training" (Stutz. et. al) where the model parameters are changed similar to WR-CP, on top of an algorithm that certifies robustness in case of adversarial noise, and reduce the inefficiency of the prediction sets. Then when applying a CP method with guarantees on top of the new model, we obtain smaller sets. This procedure seems sound as it does not break the main point of CP ( coverage guarantees) and yet reduces inefficiency. The current experimental set-up of this paper does not support their claim ( (-38%) reduction in the prediction set sizes than the worst-case approach on average), and are insufficient. Based on this, I don't believe the experiments fairly and constructively asses performance of WR-CP compared to the SOTA or the prior works.

5. very very Minor typos ~ there is no fig 1(c) , line 153 "adaption"

**Questions:**

1. With respect to point 1 in the "Weaknesses",  I am concerned about the practicality of the algorithm 1 ( WR-CP), and computational overhead. Further, could you elaborate on how alternate solutions in the existing literature can potentially solve the density-estimation problem - or be integrated within your pipeline ( if any according to your literature-review ). I believe this is a major over-head that can be addressed if the algorithm allows for alternate solutions.

2. With respect to points 2a, 2b, could you elaborate where your contributions stand with respect to the CP pipeline ( as described ).  I believe the paper can benefit from an alternative narrative where the contributions are more on the conformal-based training procedure, and formal gaurantees are provided. Providing formal gaurantees during the trainning phase might be difficult, however applying a calibration procedure with the gaurantees to the "changed" model could mitigate this issue.

3. with regard to point 2c above, I am concerned about "the fairness" in the comparison with the baselines in which coverage is gauranteed.

4. I believe the related works should include conformal training based procedures where the model parameters are explicitly changed in order to optimize a conformal-based efficiency, in order to clearly place your contributions with respect to the literature. Further, the application to Multi-source conformal prediction calls for a more thorough literature review in this context. there are works that use a similar framework and address distribution shift in a multi-agent CP framework, or use a similar global distribution to introduce distributed/federated conformal procedures. None of those were addressed in this work.

---

> ### Author Response · Authors · 2024-11-20
> **Responses to weakness 1, 2(a), and related questions.**
>
> Thank you for your insightful comments and questions! We appreciate your feedback and would like to provide further clarification on our work. We consolidated several comments regarding the weaknesses and questions into a single response and addressed them collectively. References are listed at the end of the series of responses. Please note that we have uploaded a **revised version**, which includes additional details for your review.
> >The minimization of the term corresponding to covariate shift, seems to just use previous contributions in the field (Tibshirani et. al 2019).
>
> Although importance weighting is proposed to minimize the coverage gap caused by covariate shifts in the previous work [13], the effectiveness of this technique under joint distribution shift has not been studied before. We proved its effectiveness under joint distribution shift using the framework of pushforward measures in Section 3.2.
>
> >With respect to point 1 in the "Weaknesses", I am concerned about the practicality of the algorithm 1 (WR-CP), and computational overhead. Further, could you elaborate on how alternate solutions in the existing literature can potentially solve the density-estimation problem - or be integrated witin your pipeline ( if any according to your literature-review ). I believe this is a major over-head that can be addressed if the algorithm allows for alternate solutions.
>
> Kernel density estimation (KDE) is a component of the proposed method but not the only option and this component can be replaced easily with other density estimation methods, such as nearest neighbor estimation and wavelet density estimation.
> Applying KDE to conduct importance weighting is a widely applied approach, as demonstrated in CoDrug[1] and SLCP[2].  As mentioned in **Appendix C.1**, we applied grid search to find the best bandwidth for KDE. This method may increase computation cost with repeated calculation, especially for high-dimension data. Bernacchia–Pigolotti KDE[3] derived a method for objectively determining both the kernel shape and the kernel bandwidth with a Fourier-based filter. FastKDE[4] further extended Bernacchia–Pigolotti KDE to high-dimension data to improve computation efficiency. We believe these ideas can be applied to our work to mitigate the computational overhead.
>
> >There are no formal robustness/coverage guarantees on the final prediction sets. The way the problem is introduced, this becomes a major limitation.
>
> >Providing formal gaurantees during the trainning phase might be difficult, however applying a calibration procedure with the gaurantees to the "changed" model could mitigate this issue.
>
> As you suggested, it is worth studying the coverage guarantee of the proposed WR-CP method. Under the setup of multi-source conformal prediction, with $\tau$ as the $1-\alpha$ quantile of the weighted calibration conformal score distribution $Q_{V,s_P}$, we can derive the coverage gap upper bound by
> \begin{equation}
> |F _{Q _{V,s_P}}(\tau) - F _{Q _{V}}(\tau)| = \left| \sum _{i=1}^k w _i F _{D _{V,s _P}^{(i)}}(\tau) - \sum _{i=1}^k w _i F _{D^{(i)} _{V}}(\tau) \right|\leq \sum _{i=1}^k w _i |F _{D^{(i)} _{V,s _P}}(\tau) - F _{D^{(i)} _{V}}(\tau)| \leq\sup _{i\in {1,...,k }} |F _{D^{(i)} _{V,s _P}}(\tau) - F _{D^{(i)} _{V}}(\tau)|.
> \end{equation}
> In other words, the coverage gap on a test distribution must be less or equal to the largest gap at $\tau$ on multiple training distributions. Denoting $\alpha_D=\sup _{i\in\{1,...,k\}} |F _{D^{(i)} _{V,s _P}}(\tau) - F _{D^{(i)} _{V}}(\tau)|$, the coverage lower bound is $1-\alpha-\alpha_D$. Calculating $\alpha_D$ to find the lower bound is a calibration procedure with the coverage guarantee of WR-CP. A model trained by Wasserstein-regularized loss in Eq. (19) can reduce $\alpha_D$, thus making the guarantee closer to the desired $1-\alpha$ coverage. It is important to highlight that $\alpha_D$ is adaptive to variations in test conformal score distribution $Q_V$, as evident from the equation above. This adaptivity ensures that the lower bound dynamically adjusts to different $Q_V$. A detailed discussion is provided in the updated **Appendix. B.2**.

---

> ### Author Response · Authors · 2024-11-20
> **Responses to weakness 2(b) and related questions.**
>
> > With respect to points 2a, 2b, could you elaborate where your contributions stand with respect to the CP pipeline (as described ).
>
> > I believe the paper can benefit from an alternative narrative where the contributions are more on the conformal-based training procedure, and formal guarantees are provided.
>
> Indeed, there are two pipelines to improve CP robustness to test distribution shift. The first one is to modify the post-hoc CP procedure, as exemplified by RSCP [5] and Robust Validation [14]. The second one is to embed a conformal-based loss during the training phase, like RCT [6]. These works can provide theoretical coverage guarantees with assumptions of test distribution shifts.  For instance, RSCP assumes the test distribution shift as constrained adversarial noise on features, so the label marginal distributions are identical.
>
> The first core contribution of our work is the theoretical analysis in **Sections 3 and 4** without relying on any distribution shift assumption. We focus on understanding how the difference between the test and calibration data distributions is transformed into the difference in their conformal score distributions, which in turn enlarges the coverage gap. We introduce Wasserstein distance (W-distance) to quantify the process.  An analogousness of our work is NexCP [7], which applies total variation distance to do so. We highlighted the limitations of the total variation distance in **Figure 1(b)**, with a detailed explanation in **Appendix F.1** and experimental validation in **Section 6.2**. Furthermore, while the NexCP algorithm depends on the reliability of a fixed weighting scheme to reduce total variation distance, applying our theorems to multi-source CP does not rely on it. This leads to the proposed WR-CP.
>
> The second core contribution is the WR-CP algorithm under a multi-source setup, where a test distribution is an unknown random mixture of training distributions. WR-CP achieves a desirable trade-off between coverage gap and prediction set size, which is justified in the next question. In addition, with a coverage guarantee at $1-\alpha-\alpha_D$ obtained by a calibration procedure as introduced above, WR-CP belongs to the second pipeline, since we incorporate a conformal-based loss term during training.
>
> It is necessary to clarify the benefits and distinctiveness of **WR-CP with a coverage guarantee**. Starting from a broader perspective, regardless of specific tasks (whether regression or classification) and the associated pipelines, RSCP [5], Robust Validation [14], and RCT[6] provide coverage guarantees with the worst-case principle, which outputs prediction sets exclusively relying on the upper bound of $1-\alpha$ quantile of shifted test conformal scores.  The prediction efficiency of the worst-case principle is susceptible to an 'outlier' test distribution. To give a concrete example, consider we have multiple shifted test distributions. There is an 'outlier' test distribution whose $1-\alpha$ conformal score quantile, denoted as $\tau_{max}$, is much larger than the conformal score quantiles of other test distributions. The worst-case principle generates prediction sets for samples from all test distributions with $\tau_{max}$. This will cause low prediction efficiency for samples not from the 'outlier' distribution. Consequently, CP methods guided by the worst-case principle will suffer from the susceptibility. In contrast, the guaranteed WR-CP takes another pathway to get rid of the susceptibility. WR-CP generates prediction sets based on the $1-\alpha$ quantile of importance-weighted calibration conformal scores with a modified coverage guarantee $1-\alpha-\alpha_D$. This guarantee is adaptive to different test distributions, as evidenced by the definition of $\alpha_D$, so prediction efficiency is not vulnerable to an outlier test distribution. Nevertheless, we admit that we take advantage of the multi-source setup to obtain the modified guarantee, while the worst-case principle does not rely on this setup. Secondly, compared with RCT [6] within the second pipeline, WR-CP does not require retraining when $1-\alpha$ changes, as Wasserstein-regularization does not take the confidence level as a hyperparameter.
>
> >The current experimental set-up of this paper does not support their claim ( (-38%) reduction in the prediction set sizes than the worst-case approach on average), and is insufficient.
>
> We would like to clarify that **Figure 4** aims to show that WR-CP can significantly improve prediction efficiency (38% prediction set size reduction) with only a mild loss in coverage (3.1%coverage gap), and **Figure 5** proves that this trade-off is controllable. In contrast, worst-case CP (WC-CP) does not present an alternative when efficiency (rather than guarantee) is also a priority. Specifically, under the existence of an 'outlier' test distribution, prediction sets for samples from other test distributions can be excessively large and uninformative.

---

> ### Author Response · Authors · 2024-11-20
> **Responses to weakness 2(c) and related questions.**
>
> > I do not believe the experiments are a fair comparison, as the model parameters are "changed". The fair comparison would apply a fixed calibration procedure, with guarantees to both models ( one baseline trained without any insights from the Wasserstein upper bound, one trained according to WR-CP) in order to measure the gains in terms of coverage gap or the length-inefficiency. To indicate this point further, (https://lilywenglab.github.io/Provably-Robust-Conformal-Prediction/) applied "Conformal Training" (Stutz. et. al) where the model parameters are changed similar to WR-CP, on top of an algorithm that certifies robustness in case of adversarial noise, and reduce the inefficiency of the prediction sets. Then when applying a CP method with guarantees on top of the new model, we obtain smaller sets.
>
> We understand your concerns about fairness. However, the method you suggested may not suit WR-CP. It is important to note that WR-CP in Algorithm 1 encompasses both the training and prediction phases. Applying other post-hoc CP methods, like WC-CP, on a model trained by Wasserstein-regularized loss in Eq. (19) compromises the integrity of WR-CP.
>
> We also experimentally demonstrated that the combination of a model trained by Wasserstein-regularized loss and WC-CP (called Hybrid WR-WC) can result in larger prediction sets than applying WC-CP to a residual-driven (i.e. unregularized) model.   The average prediction set sizes with coverage guarantee $1-\alpha=0.9$ are listed below, with standard deviations shown in parentheses.
> |              |   Airfoil   |    PeMSD4   |    PeMSD8   |    Seattle   |     U.S.    |     Japan    |
> |:------------:|:-----------:|:-----------:|:-----------:|:------------:|:-----------:|:------------:|
> |     WC-CP    | 1.60 (0.76) | 7.17 (0.19) | 4.46 (0.26) | 15.18 (0.34) | 7.52 (1.25) | 11.22 (0.58) |
> | Hybrid WR-WC | 2.50 (1.46) | 8.58 (0.87) | 5.50 (0.45) | 20.11 (1.83) | 8.13 (1.40) | 11.97 (1.45) |
>
> To explain the result, while regularization in Hybrid WR-WC enhances the reliability of calibration conformal score distributions, the WC component depends exclusively on the upper bound of $1-\alpha$ test conformal score quantile, rendering it unable to benefit from regularization. Even more problematically, Hybrid WR-WC results in larger prediction sets, as the regularization inevitably increases the prediction residuals, which in turn increases the upper bound of the test conformal score quantile. Please check **Appendix B.1** and **Figure 6** for details and visualized results of the experiments.
>
> >Based on this, I don't believe the experiments fairly and constructively assess the performance of WR-CP compared to the SOTA or the prior works.
>
> >With regard to point 2c above, I am concerned about "fairness" in the comparison with the baselines in which coverage is guaranteed.
>
> To further address your concerns about 'fairness', we evaluated the prediction efficiency of **WR-CP with $1 - \alpha - \alpha_D$ guarantee**. Specifically, we set $\alpha=0.1$ and computed the corresponding $\alpha_D$ for various test distributions. Additionally, we calculated the coverage and prediction set size of WC-CP on each test distribution, using the corresponding guarantee at $1 - \alpha - \alpha_D$ for comparison. The average prediction set sizes for WR-CP and WC-CP across six datasets are presented below, with standard deviations shown in parentheses. WR-CP led to a **26.9% prediction set size reduction** across six datasets on average.
> |       |   Airfoil   |    PeMSD4   |    PeMSD8   |    Seattle   |     U.S.    |    Japan    |
> |:-----:|:-----------:|:-----------:|:-----------:|:------------:|:-----------:|:-----------:|
> | WC-CP | 0.38 (0.21) | 4.96 (0.34) | 3.15 (0.34) | 11.56 (0.26) | 4.63 (1.08) | 2.45 (1.11) |
> | WR-CP | 0.41 (0.25) | 3.78 (0.77) | 2.16 (0.83) | 10.95 (0.61) | 1.91 (1.10) | 1.01 (0.26) |
>
> The table illustrates an improvement in prediction efficiency for the PeMSD4, PeMSD8, U.S.-States, and Japan-Prefectures datasets. However, the efficiency remains nearly constant for the Seattle-loop dataset and even declines for the airfoil self-noise dataset. This behavior can be explained by the regularization mechanism. Wasserstein-regularization can help create prediction sets by using the quantile of calibration conformal scores instead of relying on the upper bound of test conformal score quantiles from a residual-driven model. However, regularization can also increase prediction residuals, which may lead to a higher calibration conformal score quantile. If the benefit of regularization outweighs the drawback, the result is smaller prediction sets. Conversely, if the increase in the calibration conformal score quantile matches or exceeds the upper bound of the test conformal score quantile, the prediction sets remain the same size or become larger, respectively. We refer to the updated **Appendix B.2** and **Figure 7** for visualized experiment results.

---

> ### Author Response · Authors · 2024-11-20
> **Responses to question 4.**
>
> > I believe the related works should include conformal training-based procedures where the model parameters are explicitly changed in order to optimize a conformal-based efficiency, in order to clearly place your contributions with respect to the literature.
>
> As you mentioned, ConfTr [8] trains the model to improve prediction efficiency, but it relies on exchangeability and does not consider distribution shifts.  While RCT [6] focuses on both prediction efficiency and robustness to distribution shift, it works on classification tasks and applies the worst-case principle during training, making prediction efficiency susceptible to 'outlier' test distributions, as we discussed above. These related works are updated in **Section 2.2**.
>
> In our work, first, the theories in Section 3 and Section 4 do not assume the form in which the shift exists. Secondly, the proposed WR-CP does not suffer from the susceptibility to 'outlier' test distributions as the worst-case principle is not adopted in the algorithm.  Also, as we mentioned before, this algorithm does not require retraining when $1-\alpha$ changes.
>
> >Further, the application to Multi-source conformal prediction calls for a more thorough literature review in this context. there are works that use a similar framework and address distribution shift in a multi-agent CP framework, or use a similar global distribution to introduce distributed/federated conformal procedures. None of those were addressed in this work.
>
> A more comprehensive literature review of multi-source CP can help elaborate our contributions. Specifically, multi-source CP considers that the calibration distribution is a known mixture of several distinct subpopulations.  Within this framework, FCP [9] aims for a coverage guarantee when the test and calibration samples are from the same mixture. Similarly, FedCP-QQ [11] considers i.i.d. calibration and test samples from a uniformly weighted mixture and designs a quantile-of-quantiles estimator to aggregate the conformal score quantile of each subpopulation. When exchangeability does not hold, DP-FedCP [10] addresses scenarios where test samples are drawn from a single subpopulation, assuming that only label shifts ($P_Y\neq Q_Y$) occur among the subpopulations. Besides, CP with missing outcomes is studied in [12], while the samples from the test distribution are accessible. WR-CP works on a different setup: the test samples are drawn from an unknown random mixture where both concept and covariate shifts can occur among subpopulations. These related works are updated in **Section 2.2**.
>
> **Reference**
>
> [1] Laghuvarapu, Siddhartha, Zhen Lin, and Jimeng Sun. "CoDrug: Conformal Drug Property Prediction with Density Estimation under Covariate Shift." Advances in Neural Information Processing Systems 36 (2024).
>
> [2]Han, Xing, et al. "Split localized conformal prediction." arXiv preprint arXiv:2206.13092 (2022).
>
> [3]Bernacchia, Alberto, and Simone Pigolotti. "Self-consistent method for density estimation." Journal of the Royal Statistical Society Series B: Statistical Methodology 73.3 (2011): 407-422.
>
> [4]O’Brien, Travis A., et al. "A fast and objective multidimensional kernel density estimation method: fastKDE." Computational Statistics & Data Analysis 101 (2016): 148-160.
>
> [5]Gendler, Asaf, et al. "Adversarially robust conformal prediction." International Conference on Learning Representations. 2021.
>
> [6]Yan, Ge, Yaniv Romano, and Tsui-Wei Weng. "Provably robust conformal prediction with improved efficiency." arXiv preprint arXiv:2404.19651 (2024).
>
> [7]Barber, Rina Foygel, et al. "Conformal prediction beyond exchangeability." The Annals of Statistics 51.2 (2023): 816-845.
>
> [8]Stutz, David, Ali Taylan Cemgil, and Arnaud Doucet. "Learning optimal conformal classifiers." arXiv preprint arXiv:2110.09192 (2021).
>
> [9]Lu, Charles, et al. "Federated conformal predictors for distributed uncertainty quantification." International Conference on Machine Learning. PMLR, 2023.
>
> [10]Plassier, Vincent, et al. "Conformal prediction for federated uncertainty quantification under label shift." International Conference on Machine Learning. PMLR, 2023.
>
> [11]Humbert, Pierre, et al. "One-shot federated conformal prediction." International Conference on Machine Learning. PMLR, 2023.
>
> [12]Liu, Yi, et al. "Multi-Source Conformal Inference Under Distribution Shift." arXiv preprint arXiv:2405.09331 (2024).
>
> [13]Tibshirani, Ryan J., et al. "Conformal prediction under covariate shift." Advances in neural information processing systems 32 (2019).
>
> [14]Cauchois, Maxime, et al. "Robust validation: Confident predictions even when distributions shift." Journal of the American Statistical Association (2024): 1-66.

---

> > ### Author Response · Authors · 2024-11-22
> > **We appreciate your feedback and advices.**
> >
> > Thank you so much for your valuable feedback on this work. We implemented your suggestions and would love to hear your thoughts on whether these changes align with your expectations. Your input would be greatly appreciated to ensure the best outcome.

---

> ### Author Response · Authors · 2024-11-24
> **The discussion of KDE computation efficiency is included in Appendix E.2.**
>
> > I am concerned about the practicality of the algorithm 1 (WR-CP), and computational overhead. Further, could you elaborate on how alternate solutions in the existing literature can potentially solve the density-estimation problem - or be integrated witin your pipeline ( if any according to your literature-review ). I believe this is a major over-head that can be addressed if the algorithm allows for alternate solutions.
>
> We have added **Appendix E.2** to address the computational efficiency challenges associated with KDE and explore how existing methods can enhance performance. Furthermore, we have expanded **Appendix E.1** to provide a detailed discussion of the numerical instability in KDE. This includes an analysis of the underlying causes and the steps we have taken to alleviate the issue, such as employing a carefully selected kernel, applying feature normalization, and conducting a comprehensive grid search for optimal bandwidth selection.

---

> ### Author Response · Authors · 2024-11-25
> **Section 2.2 includes a more comprehensive literature review of CP in multi-source domain generalization and federated setting**
>
> >Further, the application to Multi-source conformal prediction calls for a more thorough literature review in this context. there are works that use a similar framework and address distribution shift in a multi-agent CP framework, or use a similar global distribution to introduce distributed/federated conformal procedures. None of those were addressed in this work.
>
> We refined the literature review of CP in multi-source domain generalization and federated settings in **Section 2.2**, making it more coherent and explicitly highlighting the differences between these two areas. Basically, CP under multi-source domain generalization focuses on developing a model that generalizes effectively to **unseen test distributions** by leveraging the data from multiple source distributions. Federated CP is a related yet distinct area, that aims to train a model across decentralized data sources to perform well on **a specific test distribution** (typically a uniformly weighted mixture of source distributions) without requiring centralization to ensure privacy. Regarding federated CP, FCP [9] and FedCP-QQ [11]  aim for a coverage guarantee when the test and calibration samples are exchangeable from the same mixture. When exchangeability does not hold,  DP-FedCP [10] and [12] explore the scenario when label shifts and missing labels occur among source distributions, respectively.  The proposed WR-CP does not consider privacy but works on a more generalized setup: the test samples are drawn from an **unknown random mixture** where both concept and covariate shifts can occur among the source domains.

---

> ### Author Response · Authors · 2024-11-30
> **We look forward to your feedback**
>
> Thank you for your thoughtful review! It really motivates us to dig deeper into our work and refine our approach. We've made some great progress in addressing your concerns, such as the coverage guarantee of the proposed WR-CP (**Appendix B.2**), the connection to federated CP (**Section 2.2**), and methods for reducing the computational cost of KDE (**Appendix E.2**). We hope the revisions we’ve made help clarify the unique aspects of our contribution. We’re eager to hear your thoughts on the changes we’ve made and look forward to your feedback!

---

> > ### Comment · Reviewer_GF9E · 2024-12-02
> >
> > I thank the authors for their response. They have clarified most of my concerns and questions. I do believe the theoretical contribution of this paper is good, and I will raise my score to a 6.

---

> > > ### Author Response · Authors · 2024-12-03
> > > **Thank You for Your Constructive Review**
> > >
> > > Thank you for your insightful review. It encourages us to explore the potential of our work further, including aspects such as the proposed algorithm's coverage guarantee. We truly appreciate the time and effort you dedicated to reviewing the paper!

---

### Official Review · Reviewer_C65Z · 2024-11-05

**Soundness:** 3
**Presentation:** 2
**Contribution:** 3
**Rating:** 8
**Confidence:** 4

**Summary:**

This paper addresses conformal prediction under general distribution shift, including both covariate and concept shift. The authors propose using the Wasserstein distance to bound the coverage gap of conformal prediction methods in shifted distributions. Then, they decompose the Wasserstein distance between calibration and test distributions into two parts which relates to the covariate shift and concept shift separately. For scenarios where test samples are available, they use the Wassertein distance between empirical calibration and test distributions to approximate the true Wasserstein distance with concentration bound. Finally, they also consider the case without samples from the test distributions, in a multi-source conformal prediction setting. In this setting, the test distribution is an unknown mixture of training distributions. In this case, the authors introduce a Wasserstein regularizer in the training loss, which is the uniform average of the Wasserstein distance between the pushforward measure of the conformal score of the entire training data and the conformal score of each compoent over that component.

**Strengths:**

(1) The paper rigorously explores conformal prediction under complex distribution shifts, going beyond covariate shift to address both covariate and concept shifts.

(2) The decomposition of the Wasserstein distance to measure and address different types of shifts is novel and provides a clearer structure for bounding the coverage gap.

(3) The optimization method proposed in this paper by minimizing the Wasserstein distance is very practical. The application of a Wasserstein regularizer in cases where test samples are unavailable (i.e., unknown mixture of training distributions) is a practical contribution, especially for real-world applications.

(3) The paper provides theoretical guarantees and concentration bounds. The authors validate their method across multiple datasets, showing significant improvements in coverage gap reduction and prediction set size.

**Weaknesses:**

(1) The Wasserstein distance upper bound on the coverage gap is decomposed into two parts. The term relates to the concepts shift, eta depends on the test distribution and is a little hard to interpret for me.

(2) In cases where some training components have smaller weights in the unknown mixture, the uniform averaging of the Wasserstein distance may potentially lead to suboptimal regularization effects.

(3) minor thing: in Line 243, there is no plot (c) in figure 1.

**Questions:**

(1) After decomposing the Wasserstein distance, the covariate shift component seems to be captured by the Lipschitz constant of the difference between the hypothesis and the training distribution. Is it correct to interpret that a smoother difference between the hypothesis and the training distribution would improve robustness against covariate shift, since intuitively, if this difference is smooth over x, then covariate shift will change the prediction very little? Also, this kappa doesn't depend on the test distribution, so it also means the covariate shift part is easier?

(2) Is there any geometric intuition on the term eta, which captures the concept shift?

(3) When the test distribution is an unknown mixture of the training distribution, the Wasserstein regularizer term seems to ask the conformal prediction has a small average wasseerstein distance over all components. Would it has negative effects on the results if some component with small weights in practice?

---

> ### Author Response · Authors · 2024-11-20
>
> Thank you for your insightful comments and questions! We appreciate your feedback and would like to provide further clarification on our work. Please note that we have uploaded a **revised version**, which includes additional details for your review.
> > After decomposing the Wasserstein distance, the covariate shift component seems to be captured by the Lipschitz constant of the difference between the hypothesis and the training distribution. Is it correct to interpret that a smoother difference between the hypothesis and the training distribution would improve robustness against covariate shift, since intuitively, if this difference is smooth over x, then covariate shift will change the prediction very little?
>
> Yes, your understanding is correct. If $\kappa$ is quite small, the output $s_P(x)$ will not change dramatically as $x$ changes. Consequently, as Eq. (12) indicates, covariate shift will not lead to a large Wasserstein distance (W-distance) between calibration and test conformal score distributions so that prediction sets will change little.
>
> >Also, this kappa doesn't depend on the test distribution, so it also means the covariate shift part is easier?
>
> Could you please clarify the meaning of "the covariate shift part is easier"? We assume you are asking that if minimizing $\kappa W(P_X,Q_X)$ is easier than minimizing $\eta W(Q_{Y,f_P},Q_Y)$. To handle the covariate-shift-induced W-distance, importance weighting is an implementable way if we know the test feature distribution. On the other hand, minimizing $\kappa$ may not be an easy task. Although $\kappa$ is not related to $s_Q(x)=|h(x)-f_Q(x)|$, it depends on the sensitivity of $s_P(x)=|h(x)-f_P(x)|$ to $x$. Usually, a mode $h$ does not lead to a small $\kappa$ without explicitly being trained with a $\kappa$-minimizing loss.
>
> >Is there any geometric intuition on the term eta, which captures the concept shift?
>
> To provide a geometric intuition of $\eta$, we expand the definition of $\eta$ as
> \begin{equation}
>         \eta= \max\limits_{x_1,x_2\in\mathcal{X}}\frac{|s_P(x_1)-s_Q(x_2)|}{|f_P(x_1)-f_Q(x_2)|}=\max\limits_{x_1,x_2\in\mathcal{X}}\frac{|s\left(x_1,f_P(x_1)\right)-s\left(x_2,f_Q(x_2)\right)|}{|f_P(x_1)-f_Q(x_2)|}=\max\limits_{x_1,x_2\in\mathcal{X}}\frac{\left||h(x_1)-f_P(x_1)|-|h(x_2)-f_Q(x_2)|\right|}{|f_P(x_1)-f_Q(x_2)|}.
> \end{equation}
> We first simplify the definition by assuming $x_1=x_2$, so the denominator is the absolute difference between two ground-truth mapping functions $f_P$ and $f_Q$ at $x_1$, and the numerator is the absolute difference of the residuals of $f_P$ and $f_Q$ with a given model $h$ at $x_1$. $\eta$ is the largest ratio between the two absolute differences. A small $\eta$  means even if $f_P$ and $f_Q$ differ significantly, $h$ results in similar prediction residuals on $f_P$ and $f_Q$. When $x_1\neq x_2$, $\eta$ is the largest ratio of the two absolute differences at two positions, $x_1$ and $x_2$, so a small $\eta$ means that $h$ can lead to similar residuals when $f_P(x_1)$ and $f_Q(x_2)$ differ. The expanded definition above includes both $x_1=x_2$ and $x_1\neq x_2$ conditions. Intuitively, the residual difference caused by concept shift will be constrained by $\eta$.  We also provide a visualized example to present the geometric intuition in **Appendix F.2** via **Figure 20**.
>
> >When the test distribution is an unknown mixture of the training distribution, the Wasserstein regularizer term seems to ask the conformal prediction has a small average wasseerstein distance over all components. Would it has negative effects on the results if some component with small weights in practice?
>
> Yes, your concern is reasonable. If we can forecast that the test distribution may have higher weights for some specific training subpopulations, we can proportionally increase the corresponding weights in the regularization term during training to have a more targeted model. Also, the weights $w_i$ in line 349 can be substituted with known weight values to obtain a more precise upper bound. Without such a strong assumption, we follow the maximal entropy principle and assume that each sub-population has the same weight.

---

> > ### Author Response · Authors · 2024-11-22
> > **We appreciate your feedback and advices.**
> >
> > Thank you for taking the time to provide such thoughtful feedback on our work. We sincerely hope that our responses, along with the additional clarifications in the appendix, have addressed your questions. We would be grateful for any further comments or suggestions to help us enhance the quality of this work.

---

> > > ### Comment · Reviewer_C65Z · 2024-11-30
> > >
> > > Thank the authors for detailed responses, which answer my questions. I will maintain my score.

---

> > > > ### Author Response · Authors · 2024-11-30
> > > > **Thank You for Your Insightful Review**
> > > >
> > > > Thank you so much for your insightful review! Your questions about $\kappa$ and $\eta$ motivate us to investigate further how the parameters control the Wasserstein distance between calibration and test conformal scores. These questions indeed help us improve the quality of our work. We really appreciate your efforts in reviewing the paper!

---

### Meta-Review · Area_Chair_vwd3 · 2024-12-24

**Metareview:**

The paper introduces a method called Wasserstein-Regularized Conformal Prediction (WR-CP) to address the challenge of maintaining reliable coverage guarantees in conformal prediction under joint distribution shifts, including covariate and concept shifts. Unlike previous approaches using metrics like Total Variation Distance, WR-CP leverages Wasserstein distance to decompose the coverage gap into contributions from covariate and concept shifts. This decomposition allows for a more detailed understanding of how shifts affect prediction sets and enables targeted mitigation strategies. WR-CP integrates Wasserstein-regularized training and importance weighting to reduce prediction set sizes adaptively, achieving a balance between efficiency and coverage. Theoretical analysis provides finite-sample bounds on the Wasserstein-based gap, and experiments on multiple datasets demonstrate significant reductions in prediction set sizes. with only a minor loss in coverage. While the method is more adaptive and interpretable than worst-case approaches, its theoretical guarantees are less tight, and the computational overhead of Wasserstein distance estimation can be significant.

Novel Decomposition of Coverage Gaps and practical Impact and Adaptability seems innovative and I would recommend an accept.
I also recommend the authors to simplify some of the notations, and reduce the clutter at several points

**Additional Comments On Reviewer Discussion:**

During the review discussion, reviewers acknowledged the novelty and theoretical contributions but expressed concerns about certain aspects. Specifically, they questioned the fairness of experimental setups, where model parameters were changed, the computational overhead of KDE, and the absence of formal coverage guarantees. The authors addressed these points by clarifying the benefits of Wasserstein distance, discussing numerical stability in KDE, and providing additional experiments that compared WR-CP to baselines. They also refined the literature review to better situate WR-CP within the context of multi-source and federated conformal prediction. Despite residual concerns about the practicality of the proposed method and its comparative fairness, reviewers recognized the strong theoretical contributions, ultimately converging on a positive but cautious assessment of the work.

---

### Decision · Program_Chairs · 2025-01-22

Accept (Poster)